

# Novel Aerosol Flow Reactor to Study Secondary Organic Aerosol

Kelly L. Pereira[1], Grazia Rovelli[2,3], Young C. Song[2], Alfred W. Mayhew[1], Jonathan P. Reid[2], Jacqueline F. Hamilton[1]

[1]Wolfson Atmospheric Chemistry Laboratories, Department of Chemistry, University of York, York, YO10 5DD, UK
[2]School of Chemistry, Cantock's Close, University of Bristol, Bristol, BS8 1TS, UK
[3]Lawrence Berkeley National Laboratory, Chemical Sciences Division, Berkeley, CA 94720, USA

*Correspondence to:* Jacqueline Hamilton (jacqui.hamilton@york.ac.uk)

## Abstract

Gas-particle equilibrium partitioning is a fundamental concept used to describe the growth and loss of secondary organic aerosol (SOA). However, recent literature has suggested that gas-particle partitioning may be kinetically limited, preventing volatilization from the aerosol phase as a result of the physical state of the aerosol (*e.g.* glassy, viscous). Experimental measurements of diffusion constants within viscous aerosol are limited and do not represent the complex chemical composition observed in SOA (*i.e.* multicomponent mixtures). Motivated by the need to address fundamental questions regarding the effect of the physical state and chemical composition of a particle on gas-particle partitioning, we present the design and operation of a newly built 0.3 m$^3$ continuous flow reactor (CFR) which can be used as a tool to gain considerable insights into the composition and physical state of SOA. The CFR was used to generate SOA mass from the photo-oxidation of α-pinene, limonene, β-caryophyllene and toluene under different experimental conditions (*i.e.* relative humidity, VOC and VOC/NO$_x$ ratios). Up to 10$^2$ mg of SOA mass was collected per experiment, allowing the use of highly accurate compositional and single particle analysis techniques which are not usually accessible, due to the large quantity of organic aerosol mass required for analysis. A suite of offline analytical techniques was used to determine the chemical composition and physical state of the generated SOA, including: attenuated total reflectance infra-red spectroscopy, CHNS elemental analyser, $^1$H and $^1$H-$^{13}$C nuclear magnetic resonance spectroscopy (NMR), ultra-performance liquid chromatography ultra-high resolution mass spectrometry (UHRMS), high performance liquid chromatography ion-trap mass spectrometry (HPLC-ITMS) and an electrodynamic balance (EDB). The oxygen-to-carbon (O/C) and hydrogen-to-carbon (H/C) ratios of generated SOA samples (determined using a CHNS elemental analyser) displayed very good agreement with literature values and were consistent with the characteristic Van Krevelen diagram trajectory, with an observed slope of -0.41. The elemental composition of two SOA samples formed in separate replicate experiments displayed excellent reproducibility, with the O/C and H/C ratios of the SOA samples observed to be within error of the analytical instrumentation (instrument accuracy ± 0.15 % to a reference standard). The ability to use a highly accurate CHNS elemental analyser to determine the elemental composition of the SOA samples, allowed us to evaluate the accuracy of reported SOA elemental compositions using UHRMS (a commonly used technique). In all of the experiments investigated, the SOA O/C ratios obtained for each SOA sample using UHRMS were lower than the O/C ratios obtained from the CHNS analyser (the more accurate and non-selective technique). The average difference in the



ΔO/C ratios ranged from 19 to 45 % depending on the SOA precursor and formation conditions. α-pinene SOA standards were generated from the collected SOA mass using semi-preparative HPLC-ITMS coupled to an automated fraction collector, followed by [1]H NMR spectroscopy. Up to 35.8 ± 1.6 % (propagated error of the uncertainty in the slope of the calibrations graphs) of α-pinene SOA was quantified using this method; a considerable improvement from most previous studies. Single

aerosol droplets were generated from the collected SOA samples and trapped within an EDB at different temperatures and relative humidities to investigate the dynamic changes in their physiochemical properties. The volatilisation of organic components from toluene and β-caryophyllene SOA particles at 0 % relative humidity was found to be kinetically limited, owing to particle viscosity. The unconventional use of a newly-built CFR combined with comprehensive offline chemical characterisation and single particle measurements, offers a unique approach to further our understanding of the relationship/s

between SOA formation conditions, chemical composition and physiochemical properties.

## 1. Introduction

Organic aerosol (OA) accounts for a substantial fraction of ambient particulate matter (Kroll and Seinfeld (2008); Kanakidou et al. (2005) and references therein) and exhibits substantial chemical complexity. OA contains thousands of compounds of differing chemical functionalities, volatilities and masses (Goldstein and Galbally, 2007; Kanakidou et al., 2005; Seinfeld and

Pankow, 2003). This chemical complexity poses a significant analytical challenge (Goldstein and Galbally, 2007; Nozière et al., 2015; Zhang et al., 2011). OA can be broadly characterised into two sources, primary organic aerosol which is directly emitted into the atmosphere and secondary organic aerosol (SOA) which is formed in the atmosphere from the oxidation of volatile organic compounds (VOCs). SOA formation mechanisms are poorly understood. It has been estimated that $10^4$ to $10^5$ VOCs are present in the atmosphere (Goldstein and Galbally, 2007). These VOCs can undergo numerous oxidation reactions,

forming lower volatility compounds which can partition into the particulate phase forming SOA. Once formed, the chemical composition of SOA can continue to evolve *via* absorptive partitioning (Donahue et al., 2006; Pankow, 1994a, b), reactive uptake of organic gases (Czoschke et al., 2003; Gaston et al., 2014; Iinuma et al., 2004; Jang et al., 2002; Kroll et al., 2005; Riva et al., 2017), heterogenous oxidation by gas-phase radicals (George and Abbatt, 2010a; George and Abbatt, 2010b; Kroll et al., 2015; Li et al., 2018) and in-particle phase reactions (Gao et al., 2004; Heaton et al., 2007; Kalberer et al., 2004; Tolocka

et al., 2004).

The chemical and physical transformations of SOA are often studied using atmospheric simulation chambers or oxidative flow reactors (*e.g.* (Bloss et al., 2005; Cocker et al., 2001; Friedman and Farmer, 2018; Glowacki et al., 2007; Hamilton et al., 2011; Hodshire et al., 2018; Liu and Zeng, 2018; Liu et al., 2018; Rohrer et al., 2005; Witkowski et al., 2018)). These techniques

allow SOA formation and ageing to be investigated under controlled conditions. The oxidation of a single VOC can be investigated, reducing atmospheric complexity and providing greater insights into SOA formation mechanisms. Atmospheric simulation chambers and oxidative flow reactors have several fundamental differences. Atmospheric simulation chambers





range in size from ~ 0.01 to 250 m$^3$ (Lambe et al., 2011). Reactants are typically introduced at the beginning of an experiment and the chemistry allowed to reach completion over several hours. VOC and oxidant mixing ratios are typically selected to simulate ambient conditions as close as possible, whilst ensuring sufficient gaseous and particulate phase concentrations are obtained for measurement. In contrast, oxidative flow reactors are smaller in size, ranging from ~ 0.001 to 0.01 m$^3$ (Lambe et

al., 2011). Oxidative flow reactors are operated with the continuous introduction of reactants and high oxidant mixing ratios, simulating several days of chemistry in a few seconds or minutes (Bruns et al., 2015; Lambe et al., 2011). Atmospheric simulation chambers and oxidative flow reactors are the most advanced tools for reducing atmospheric complexity and elucidating the chemical and physical transformations of SOA which occur in the atmosphere. However, despite the use of these techniques, deducing the detailed chemical speciation of SOA is a formidable analytical challenge. There is no single

analytical technique capable of providing the detailed chemical speciation of SOA with complete molecular characterisation (Hallquist et al., 2009). As a result, multiple complementary analytical techniques are often used to investigate the chemical composition of SOA. However, many of these techniques are often hindered by the lack of authentic standards, further compounding the molecular identification and quantification of SOA components.

To accurately predict SOA mass loadings in the ambient atmosphere, the chemical and physical properties driving the dynamic mechanisms occurring during SOA formation and ageing must be understood. Conventionally, particles were assumed to exist as liquids, forming an instantaneous reversible equilibrium with the gas phase, as described by the gas-particle partitioning theorem (Donahue et al., 2006; Pankow, 1994a, b). However, studies have shown that physical state of a particle can drastically reduce semi-volatile evaporation rates and in some cases, prevent evaporation (Grieshop et al., 2007; Perraud et al., 2012;

Shiraiwa et al., 2013; Vaden et al., 2011). These kinetic limitations are driven by particle viscosity, which is influenced by temperature (Koop et al., 2011; Zobrist et al., 2008), relative humidity (Bateman et al., 2015; Mikhailov et al., 2009) and the chemical composition of the particle (Bateman et al., 2015; DeRieux et al., 2018; Grieshop et al., 2007; Huang et al., 2018; Kidd et al., 2014; Koop et al., 2011; Perraud et al., 2012; Reid et al., 2018; Roldin et al., 2014; Rothfuss and Petters, 2017b; Vaden et al., 2011). Recent studies have suggested that particle viscosity is influenced by certain chemical components within

the SOA, such as oligomers (Huang et al., 2018) and nitrate containing species (Perraud et al., 2012).

The aim of this work, was to develop a methodology capable of furthering our understanding of the chemical and physical properties driving SOA transformation processes. Here, we describe the design and unconventional use of a newly built 0.3 m$^3$ continuous flow reactor (CFR). VOCs and oxidants are continuously introduced into the reactor and sample air extracted,

operating under steady-state flow conditions. In contrast to oxidative flow reactors and atmospheric simulation chambers, CFRs allow atmospheric oxidation to be simulated under stable conditions through the control of reactant mixing ratios and flow rates (residence time) (Zhang et al., 2018). The use of CFRs within aerosol science is still very much in its infancy. CFRs have been used to study the gas-phase chemistry of isoprene (Liu et al., 2013; Zhang et al., 2018), SOA formed from the ozonolysis of α-pinene (Shilling et al., 2008) and in animal toxicology, to investigate the biological effect of exposure to SOA



formed from the photo-oxidation of gasoline exhaust emissions (Papapostolou et al., 2011). In this study, the CFR was used to generate considerable quantities of SOA mass from the photo-oxidation of α-pinene, limonene, β-caryophyllene and toluene under different experimental conditions. The SOA mass collected in each experiment was analysed using highly accurate compositional and single particle analysis techniques, only possible because of the large amount of SOA mass that can be

generated by the CFR. Preliminary compositional and single particle analysis data is presented, demonstrating the capabilities of the CFR. We show how the unconventional use of a newly-built CFR combined with comprehensive offline chemical characterisation and single particle measurements, offers a unique approach to further our understanding of the relationship/s between SOA formation conditions, chemical composition and physiochemical properties.

## 2. Methods

### 2.1 Continuous Flow Reactor Design

The CFR is located in the University of York, Wolfson Atmospheric Chemistry Laboratories. The CFR consists of a 0.3 m³ custom-made polyvinyl fluoride (PVF) sampling bag with the following dimensions, 1.5 m (l) × 1.0 m (w) × 0.2 m (h) (Adtech, Gloucester, UK). The sampling bag is enclosed within a rectangular aluminium housing and supported by a ¾" stainless steel hanging rail mounted inside the enclosure. The enclosure is constructed out of Bosch Rexroth rail (Bosch Rexroth, Cambridge,

UK) with aluminium sheets covering each facia. A removable front facia panel provides access to the reactor. A schematic of the CFR is shown in Figure 1.

### 2.1.1 Reactor

The CFR has four inlets, three outlets and two septum seals. The two septum seals allow a UV pen-ray and coupled temperature and humidity sensor to be mounted inside the sampling bag. UV radiation is generated using a Hg pen-ray (UVP, Cambridge,

UK) which emits at 254 nm (primary energy) and 185 nm , forming ozone and hydroxyl radicals (in the presence of nitrous acid or water vapour) inside the reactor. The UV pen-ray can become hot to touch after serval hours of operation. Approximately 60 cm of 1/16" stainless steel tubing was wrapped around the UV pen ray in cone shape, preventing the UV lamp from touching the sampling bag without blocking the light source. The reactor temperature and humidity are measured using a TE HTM2500LF sensor (Future Electronics, Surrey, UK) operated *via* a U3 LabJack (LabJack, Colorado, USA) and

an in-house built program in DAQfactory Express software. The sampling bag is covered in foil to maximise light intensity inside the reactor and minimise ozone formation inside the enclosure. Mass flow controllers are used to balance the in and out flows of the reactor, ensuring the reactor volume remains constant. Humidified air, nitric oxide and one or two VOC precursors (*i.e.* mixed VOC precursor experiments) can be continuously introduced into the CFR *via* three separate inlets (see Figure 1). The fourth inlet is used to rapidly fill the reactor with dry purified air prior to an experiment. Humidified air is generated by

flowing dry purified air through a water bubbler. VOCs are introduced into the CFR by flowing a low flow rate (< 300 ml/min) of dry purified air through a temperature controlled glass bulb, containing the VOC precursor in a liquid state and in excess.



An additional flow of dry purified air (*i.e.* make-up flow) is used to rapidly transport the VOC vapour into the CFR *via* a heated stainless steel line, minimising condensational losses of the VOC precursor. Gaseous and particulate phase measurements can be obtained *via* any of the three reactor outlets.

### 2.1.2 Inlet System

The reactor is supplied with dry hydrocarbon scrubbed air from an oil-free compressed air generator (SPRST, Spiral air, Oxfordshire, UK) using an in-built water and in-line hydrocarbon trap (GateKeeper GPU gas purifier, Entegris, Billerica, USA). Dry hydrocarbon scrubbed air is delivered to the inlet system through ¼" perfluoroalkoxy (PFA) tubing at a pressure of ~ 110 psi (output of compressed air generator). Five ¼" PFA lines are connected to the main feed of dry hydrocarbon scrubbed air. These lines are: (i) VOC 1 (ii) VOC 2 (iii) make-up flow, (iv) humidity, and (v) fast fill (see Figure 1). The

operation of each line is controlled by a ¼" two-way tap. VOCs are introduced into the reactor through lines (i) and (ii).

A series of safety features have been installed in lines (i) and (ii) to ensure the upstream glassware is not subjected to high positive pressure. Immediately after the two-way tap, the tubing size of lines (i) and (ii) are reduced from ¼" to ⅛" PFA. The pressure and flow rate in these lines are controlled to < 8 psi and < 300 ml/min using a 3 to 7 bar pressure regulator (product

code 122-649, IMI Norgren, RS Components, Northants, UK) and a 0.005 to 1 L/min mass flow controller (Alicat, Premier Control Technologies, Norfolk, UK), respectively; at ~ 10 psi the upstream glassware over pressurises and the gas-tight seals fail. VOC precursors are contained in 250 ml three necked glass bulbs (271-1513, VWR International, Leicestershire, UK). Suba-seals (Sigma Aldrich, Dorset, UK) are used to cover each opening of the glass bulb, providing a gas tight seal. Two pieces of ⅛" stainless steel tubing, filed at one end to create a sharp point, are inserted through the rubber gas-tight suba seals

on either end of the glass bulb, allowing dry hydrocarbon scrubbed air to pass through the glassware. The suba-seal on the middle neck of the glass bulb acts as a pressure relief (in the event of over pressuring) and is left unclamped with no attached tubing. Heated stainless steel lines (heated to ~ 70°C) are used from the outlet of the VOC bulb to the inlet of the CFR to minimise condensational losses of the VOC precursor. A non-return valve is installed directly after the VOC bulb on lines (i) and (ii), see Figure 1. The non-return valves ensure the higher pressure and flow rate of the dry hydrocarbon scrubbed air in

the make-up flow line does not enter lines (i) and (ii) which would result in the VOC glass bulbs over pressurising. The VOC bulbs on lines (i) and (ii) can be temperature controlled using a water or ethylene glycol bath (temperature range of -10 to 95 °C, Grant, Cambridge, UK) or an electrothermal heater (temperature range of ambient to 450 °C, EM series, Cole-Parmer, Cambridge, UK), allowing greater control over the desired mixing ratio of the VOC precursor/s injected into the CRF (*e.g.* low volatility VOCs can be heated to introduce higher mixing ratios into the reactor).

The makeup flow line consists of ¼" PFA tubing with a pressure regulator, mass flow controller and ¼" 4-way tee, connecting the makeup flow with lines (i) and (ii) to the inlet of CFR (see Figure 1). The pressure in this line is controlled to ~ 20 psi; a sufficient pressure to achieve the full flow range on a 0.1 to 20 l/min mass flow controller (Alicat, Premier Control



Technologies, Norfolk, UK). The makeup flow has two functions, to rapidly transport the VOC precursor into the CFR and to balance the total outflows of the reactor. All other introduction lines except for the fast fill (not used during an experiment), require a set flow rate for a desired mixing ratio. The makeup flow effects overall dilution but does not require a set flow rate, allowing the flow to be increased or decreased as desired, to counter balance the outflows of the reactor. The remaining two

lines connected to the main feed of dry hydrocarbon scrubbed air are the humidity (iv) and fast fill (v) lines. The humidity and fast fill lines are connected to two separate reactor inlets (second and third inlet, see Figure 1). The humidity line controls the amount of water vapour introduced into the reactor and is comprised of ¼" PFA tubing with a pressure regulator, mass flow controller and a 1000 ml water bubbler (pyrex glass bottle with an in-house built screw top water bubbler attachment). The humidity line requires a slightly higher line pressure than the VOC lines due to the backpressure created by flowing through

the water bubbler. The pyrex glass bottle has thicker walls than the VOC glass bulbs, allowing a slightly higher line pressure to be used. The pressure in the humidity line is controlled to below ~ 10 psi; the lowest possible pressure to achieve the required flow rate. A maximum flow rate of ~ 12 l/min was used in the humidity line and was controlled using a 0.1 to 20 l/min mass flow controller (Alicat, Premier Control Technologies, Norfolk, UK). Relative humidity in the reactor can be controlled by changing the ratio of humidity line flow rate/total reactor flow rate (linear relationship). The maximum relative humidity which

could be achieved in the reactor was ~ 60%. The fast fill line is used to rapidly fill the reactor with dry hydrocarbon scrubbed air. The fast fill line pressure and flow rate are unregulated (*i.e.* output of compressed air generator), allowing the reactor to be rapidly filled when required. The pressure and flow rate of the fast fill line can be loosely controlled using the ¼" two-way tap. The final introduction line which is not connected to the main feed of dry hydrocarbon scrubbed air and subsequently not discussed above, is the nitric oxide line (see Figure 1). A standard of nitric oxide (5 or 60 ppm in $N_2$, BOC, UK) can be

introduced into the CFR through the fourth reactor inlet using a 0.02 to 2 l/min mass flow controller (Alicat, Premier Control Technologies, Norfolk, UK). The nitric oxide cylinder is easily interchangeable, allowing other oxidants or scavengers to be introduced into the reactor if required.

### 2.1.3 Outlet System

Gaseous and particulate phase measurements can be obtained *via* any of the three reactor outlets. A variety of instruments can

be coupled to the CFR and easily interchanged. The limiting factor of coupling multiple instruments to the CFR is the total reactor outflow. The higher the total reactor outflow, the more difficult it becomes to balance the reactor volume. At 30 l/min, the reactor volume is replaced every 10 minutes. Mass flow controllers considerably reduce the difficulty in balancing the reactor volume when installed on the in and out flows of the reactor. However, over several hours of operation, the reactor volume can increase or decrease due to the permeability of the PVF sampling bag and the error in the accuracy of numerous

mass flow controllers. Quick changes can be made to reactor volume by shutting off all the in or out flows to the reactor for ~ 1 to 2 minutes, decreasing or increasing the reactor volume, respectively.



Several instruments have been successfully coupled to the CFR. These include: a selected ion flow tube mass spectrometer (SIFT-MS, SYFT Technologies, New Zealand) for the measurement of VOC mixing ratios and gaseous oxidation products, an electrical low pressure impactor (ELPI, model ELPI+, Dekati, Finland) for SOA collection and real-time particle mass, number and diameter measurements, and a $NO_x$ (model 42i) and $O_3$ (model 49i, Thermo Scientific, Warrington, UK) analyser
for the measurement of $NO_x$ and $O_3$ mixing ratios, respectively. The measurements from some of these instruments are discussed below.

## 2.2 Experimental design and CFR operation

The CFR was designed to generate larger quantities of SOA mass than achieved in most studies for offline chemical composition and single particle analysis. Single particle measurement techniques, such as the electrodynamic balance and
aerosol optical tweezers, can provide information on the morphology, hygroscopicity and phase behaviour of SOA with unprecedented accuracy (see Kreiger et. al. (2012) for further information). These techniques allow the effect of environmental changes on the microphysical state of the SOA to be investigated in controlled laboratory conditions, allowing the fundamental processes governing gas-particle partitioning to be better understood. These techniques, however, require considerable quantities of SOA mass for ease of transfer to particle generators  (>20 mg of SOA per experiment). To achieve this quantity
of SOA mass, high mixing ratios (*i.e.* ppmv) of a VOC precursor and oxidant/s must be continuously introduced into a reactor over several hours of operation. The CFR is ideally suited for this application in comparison to larger and well-established atmospheric simulation chambers, due to the ability to quickly and easily clean the reactor lines and replace the sampling bag at minimal cost (~ £400).

A series of experiments were performed in the CFR to investigate the composition and physical state of the SOA formed from the photo-oxidation of α-pinene (purity 98%, Sigma Aldrich, Dorset, UK), limonene (99%) β-caryophyllene (98.5%) and toluene (99.9%) under different experimental conditions. These VOCs were selected as they include biogenic and anthropogenic emissions and well-studied VOC systems, such as α-pinene. The experimental descriptions and reactor operating conditions can be found in Table 1. The experiments were designed to systematically characterise the effect of
individual and combined experimental conditions on the composition and physical state of the SOA formed. The experiments investigated the effect of: (i) relative humidity, (ii) VOC/$NO_x$ ratios, (iii) combined effect of relative humidity and VOC/$NO_x$ ratios, and (iv) VOC mixing ratios (α-pinene experiments only). Reactants were continuously introduced into the CFR and the air sampled for real-time measurements and SOA collection. α-pinene, limonene and toluene were introduced into the reactor at room temperature (temperature controlled laboratory, ~ 21 °C). β-caryophyllene was heated to 90 °C in the VOC bulb using
an electrothermal heater. The high mixing ratios (ppmv levels) used in this study could not be measured using the SIFT-MS or $NO_x$ analyser due to the risk of instrument/detector saturation. Instead, separate experiments were performed to measure α-pinene and NO at lower mixing ratios, allowing an estimated/measured mixing ratio correction to be calculated. This correction ratio was then applied to all estimated VOC and NO mixing ratios, respectively. VOC mixing ratios were estimated from the





pure       component       vapour       pressures       calculated       using       an       online       property       prediction       tool (http://umansysprop.seaes.manchester.ac.uk/), taking into account reactor dilution and a factor correction for estimated *vs.* measured mixing ratios (based on α-pinene). A small source of $NO_x$ (primarily $NO_2$, less than ~ 20 ppbv) was present in all experiments from the dry hydrocarbon scrubbed air and also potentially from the photolysis of HONO present on the chamber walls (Carter et al., 1981, 1982; Sakamaki et al., 1983). A 5 ppm standard of NO in $N_2$ (BOC, UK) was used in low α-pinene mixing ratio experiments (exp. 7 and 14, see Table 1). In all other $NO_x$ experiments, a 60 ppm standard of NO in $N_2$ (BOC, UK) was used. An insufficient amount of SOA mass was generated from the photo-oxidation of toluene using the described CFR setup and operation. Several changes were made to increase SOA formation. These include, (i) changing the UV light source to a longer Hg pen-ray (product code = 90-0004-01, UVP, California, USA) and only investigating SOA formation at 55 % relative humidity to increase ˙OH radical formation and (ii) combining the SOA mass formed in two replicate experiments for each experimental condition investigated (see Table 1).

Each time a new VOC precursor was investigated, a new sampling bag was installed and the reactor lines and components were cleaned. The sampling bag was thoroughly cleaned before use by introducing humidified hydrocarbon scrubbed air (~ 50 % relative humidity) into the reactor under UV irradiation for ~ 2 days. Upon completion, the reactor was flushed with dry hydrocarbon scrubbed air before performing a chamber background experiment (see Table 1). SOA mass and number concentrations in the chamber background experiments ranged from 21 to 369 µg m$^{-3}$ and 1.5 to $4.7 \times 10^6$ particles/cm$^3$, respectively. At the start of each experiment, the reactor was filled to volume with dry hydrocarbon scrubbed air using the fast fill line. The reactor was then flushed with humidified hydrocarbon scrubbed air for approximately 1 ½ hours before setting the flow rates of the desired experimental relative humidity and (where applicable) nitric oxide mixing ratios. Water vapour and nitric oxide were continuously introduced into the reactor for ~ 30 minutes before introducing the VOC precursor. After ~ 25 minutes of continuous VOC introduction, the UV light was switched on. Gaseous reactants were mixed in the CFR through flow and diffusion only. High flow rates (~ 12 lpm$^{-1}$) were used in all of the experiments to accelerate mixing. Steady-state mixing ratios were achieved in the CFR after ~ 50 minutes of continuous introduction of nitric oxide and ~ 25 minutes for the VOC precursor (based on α-pinene). The reactor was cleaned at the end of each experiment by introducing humidified hydrocarbon scrubbed air (~ 50% relative humidity) into the reactor with the UV lamp on. The ozone generated from UV irradiation was left inside the reactor overnight, flushing with humidified hydrocarbon scrubbed air the next morning.

Reactor temperature, relative humidity and particle diameter, number and mass measurements were recorded every second in real-time from the start to the end of each experiment. Background particle  diameter, number and mass measurements were obtained during the continuous infusion of the water vapour and (where applicable) nitric oxide. SOA was collected using both an ELPI and onto pre-conditioned 47 mm quartz fibre filters. The reactor was used as a tool to generate large quantities of SOA mass. Subsequently, no corrections have been made for gaseous or particulate phase wall losses in this work. The ELPI was used in all the experiments shown in Table 1. The use of the SIFT-MS, $NO_x$ and $O_3$ analysers depended on their availability



with other projects, although were primarily used in the α-pinene experiments. The calibration of these instruments is discussed in the SI. Quartz fibre filters were pre-conditioned in a furnace at 550°C for 5 hours to remove any volatiles before use. The ELPI collects particles with a size range of 0.006 to 10 µm onto size specific foil lined impactor plates. The SOA collected from the ELPI in each experiment, was transferred from all the foil lined impactor plates (non-size specific) into two clean

glass vials and weighed. One vial was kept at the University of York for compositional analysis and the second vial shipped in dry ice (-80 °C) to the University of Bristol for single particle measurements. The impactor foils were replaced and the impactor cleaned with methanol and water prior to each experiment. All SOA samples were wrapped in foil to minimise photolysis degradation and stored in a freezer at -20°C until analysis.

## 2.3 SOA Chemical Characterisation

SOA samples were analysed using an extensive range of single particle and compositional techniques including: an electrodynamic balance, ultra-performance liquid chromatography ultra-high resolution mass spectrometry (UPLC-UHRMS), $^1$H and $^1$H – $^{13}$C nuclear magnetic resonance spectroscopy (NMR), attenuated total reflectance Fourier transform infra-red spectroscopy (ATR-FTIR) and a CHNS elemental analyser. In addition to the above techniques, high performance liquid chromatography ion trap mass spectrometry with a semi-preparative column and an automated fraction collector was used to

generate SOA standards to quantify individual components in α-pinene SOA. The SOA mass collected from the ELPI was used for all of the above techniques, except for ATR-FTIR spectroscopy, where the SOA collected onto quartz fibre filters were used. The ELPI collects SOA onto foil lined impactor plates, minimising potential artefacts often associated with the collection of SOA onto porous substrates (*e.g.* filters), thereby eliminating time consuming extraction processes. The ATR attachment on the FTIR spectrometer however, allows SOA filters to be directly analysed without requiring extraction.

Subsequently, SOA filters were used for this technique. The sample preparation methods, instruments and their operating parameters are discussed below.

### 2.3.1 Generating SOA standards

Individual compounds in the SOA were isolated and collected using semipreparative high performance liquid chromatography ion-trap mass spectrometer (HPLC-ITMS) coupled to an automated fraction collector (FC 203B, Gilson, Dunstable, UK); an

extension of the work performed by Finessi et al. (2014). An Agilent 1100 series HPLC (Berkshire, UK) and Bruker Daltonics HTC Plus ITMS (Bremen, Germany) were used. The semipreparative column was a reverse phase Ascentis 150 cm × 10 mm, with a 5 µm particle size (Sigma Aldrich, Dorset, UK). The HPLC solvents consisted of water (LC-MS grade, optima, Fisher Scientific, UK) with 0.1 % (v/v) formic acid (Sigma Aldrich, UK) (A) and methanol (B) (LC-MS grade, optima, Fisher Scientific, UK). Gradient elution was used, starting at 87% (A), decreasing to 27% (A) at 92 minutes, re-equilibrating to 87%

(A) at 97 minutes. A 5 minute pre-run consisting of the starting mobile phase conditions was performed prior to each injection. The flow rate was set to 3.5 ml/min with an injection volume of 100 µL. The eluent post column was split *via* a tee piece and two lengths of peek tubing to the ITMS and the automated fraction collector. The use of a narrower internal diameter peek



tubing to the ITMS resulted in a split ratio of 1:8 ITMS:fraction collector. Electrospray ionisation (ESI) was used with a nebuliser pressure of 70 psi, dry gas flow rate of 12 l/min and a dry gas temperature of 365 °C. The ITMS was operated in negative and positive ionisation mode with a scan range of *m/z* 50 – 600. Collision induced dissociation (CID) was used with a fragmentation amplitude of 2 V. The data was analysed using ESI Compass 1.3 for HTC/esquire version 4. The fraction

collector was pre-programed with the retention times of the target compounds and collected into 10 ml glass vials. The collected fractions were evaporated to dryness using a solvent evaporator (model V10, Biotage, Hertford, UK) and resuspended in 500 µl deuterium oxide with 0.05% (w/w) of 3-(trimethylsilyl)propionic-2,2,3,3-d$_4$ acid (TSP) (Sigma Aldrich, UK). Finally, the collected fractions were transferred into 5 mm tubes (Wilmad, Sigma Aldrich, UK) for NMR analysis.

### 2.3.2 NMR spectroscopy

The collected fractions and SOA samples were analysed using one-dimensional [1]H and two-dimensional [1]H-[13]C heteronuclear single quantum correlation (HSQC) NMR spectroscopy. A 700 MHz Bruker Daltonics Avance Neo NMR spectrometer equipped with a prodigy triple resonance cryoprobe was used. The operating temperature was set to 21°C. The [1]H-NMR spectra were acquired at 700 MHz, with a pulse sequence of 45° and a relaxation delay of 4 seconds. The number of scans was set to 640, resulting in a total runtime of 1 hour and 8 minutes. The [1]H-[13]C-NMR HSQC Bruker Daltonics pulse program used was

hsqcetgpsi2 with a time domain data size of 256 and 1024 for channels F1 and F2, respectively. The number of scans was set to 20, resulting in a total runtime of 1 hour and 40 minutes. The [1]H NMR spectra were analysed using TopSpin version 3.5 (Bruker Daltonics, Bremen, Germany). All spectra were baseline corrected and a spectral line broadening of 0.3 Hz was used. The concentration of the collected fractions was determined using the peak integrals of the internal standard (*i.e.* TSP) and the methyl group observed in the collected fractions from [1]H-NMR spectroscopy (see Finessi et al. (2014) and Bharti and Roy

(2012) for further information). The compound structure of the collected fractions was determined using a combination of techniques, including [1]H and [1]H-[13]C-HSQC NMR spectroscopy and the fragmentation patterns obtained from CID using the HPLC-ITMS and higher-energy collisional dissociation (HCD) using the UPLC-UHRMS. The [1]H-[13]C HSQC spectra were analysed using Spectrus Processor (ACD labs, Bracknell, UK) which contains an in-built carbon-hydrogen coupling prediction tool. This prediction tool provides estimated chemical shifts of carbon-hydrogen bonds in drawn chemical structures. Chemical

structures of known and structurally similar photo-oxidation products of toluene and β-caryophyllene were drawn in the Spectrus Processor software to aid in the interpretation of the spectra discussed in section 3.2.2.

### 2.3.3 Ultra-high resolution mass spectrometry

A proportion of the ELPI SOA mass collected from each experiment was transferred into a clean vial and dissolved in 50:50 methanol:water (optima, LC-MS grade, Fisher Scientific, UK). The α-pinene standards and SOA samples were analysed using

ultra-performance liquid chromatography with a UV/Vis detector (Dionex 3000, Thermo Scientific, Warrington, UK) coupled to an ultra-high resolution mass spectrometer (QExactive Orbitrap, Thermo Scientific, Warrington, UK). An Accucore reverse phase C$_{18}$ column 100 mm × 2.1 mm with a 2.6 µm particle size was used for compound separation (Thermo Scientific,



Warrington, UK). The mobile phase consisted of water with 0.1% (v/v) formic acid (A) and methanol (B) (optima LC-MS grade, Fisher Scientific, UK). Gradient elution was used, holding at 10% (A) for 1 minute after injection, decreasing to 90% (A) at 26 minutes, followed by re-equilibration to 10% (A) at 28 minutes. A 2 minute pre-run of the starting mobile phase conditions was performed prior to each injection. The flow rate was set to 0.3 ml/min with a 2 μl injection volume. The column

temperature was controlled at 40°C and the samples were kept at 4°C in the autosampler tray during analysis. The UV/Vis wavelength ranged from 190 to 800 nm with a data collection rate of 5 Hz per second. Heated electrospray ionisation (HESI) was used, with the following parameters; sheath gas flow rate of 70 (arb.), aux gas flow rate of 3 (arb.) capillary temperature of 320°C and an aux gas heater temperature of 320°C. Spectra were acquired in negative and positive ionisation mode with a scan range of $m/z$ 85 to 750. Higher-energy collisional dissociation (HCD) was used with a stepped fragmentation amplitude

of 65 and 115 (normalised collision energy, NCE). The number of most abundant precursors selected for fragmentation per scan was set to 10, with a 3 second dynamic exclusion and an apex trigger of 2 to 4 seconds. Two authentic standards, cis-pinonic acid (purity 98%, Sigma Aldrich, UK) and pinic acid (Santa Cruz Biotechnology, Netherlands) were used for α-pinene SOA quantification. Chromatographic integration was performed using the software package Freestyle version 1.1 (Thermo Scientific, Warrington, UK). SOA samples were also analysed using the software package Compound Discoverer version 2.1

(Thermo Scientific, Warrington, UK) which was used to extract all chromatographic peaks with a signal-to-noise ratio greater than 3 from the spectra, identify the molecular formulae and perform a mass spectral library search using $m/z$ Cloud (https://www.mzcloud.org/). Molecular formulae were calculated using the following restrictions: unlimited C, H and O atoms were allowed with a maximum of 5 N atoms. In positive ionisation mode, up to 3 Na and 2 K atoms were also allowed. Compounds with assigned molecular formulae outside of the following tolerances were excluded from the data set: oxygen-

to-carbon ratio (O/C) 0.05 to 2, hydrogen-to-carbon (H/C) ratio 0.7 to 2 (following the lower limits provided in Bateman et al. (2009)), double bond equivalent (DBE) < 20, molecular formulae accuracy < 3 ppm. Chromatographic peaks identified in the solvent or procedural blanks and the SOA sample/s were removed from the data set.

### 2.3.4 ATR-FTIR spectroscopy

SOA filter samples were analysed using Fourier transform infrared spectroscopy with a diamond attenuated total reflectance

attachment (ATR-FTIR, ALPHA, Bruker Daltonics, Bremen, Germany). The scan range was set to 500 - 4000 cm$^{-1}$ with a spectral resolution of 4 cm$^{-1}$. Spectra were acquired in absorbance with 40 scans per sample. A pre-conditioned blank filter was used as the background measurement correcting for any instrument drift during analysis (*c.f.* (Coury and Dillner, 2009)). The background measurement was subtracted from the sample spectra. The crystal was cleaned with isopropanol prior to the analysis of each sample or background measurement. Three replicate measurements were obtained for each sample. Spectral

analysis was performed using essential FTIR software (eFTIR, Madison, USA). Spectra were ATR corrected using the automated function in eFTIR software package. The use of quartz fibre filters will result in two silicon dioxide absorption peaks at wavenumbers ~ 1060 and 804 cm$^{-1}$. This region of spectra provided no additional compositional information when comparing the SOA obtained from the ELPI with the quartz fibre filter.



### 2.3.5 CHNS elemental analysis

The elemental composition of the SOA was determined using a carbon, hydrogen, nitrogen and sulfur (CHNS) elemental analyser (model CE-440, Exeter Analytical, Coventry, UK). SOA samples were weighed using a Sartorius SE2 analytical balance (Surrey, UK). The reported elemental composition consists of an average of two replicate measurements. The

remaining proportion of the SOA mass which did not contain elements C, H, N or S, has been attributed to O.

### 2.4 Single particle analysis

The confinement of single aerosol particles within optical, electrodynamic or acoustic traps, or on surfaces, allows detailed information on their chemical-physical properties and dynamics to be obtained in isolated and controlled laboratory conditions (Krieger et al., 2012). In this work, a concentric cylindrical electrodynamic balance (CC-EDB) was used to trap single SOA

particles to obtain information on some of the evolving chemical-physical properties of single SOA particles, such as their volatility distribution, possible kinetic limitations to evaporative loss of volatile species, hygroscopic and optical properties, phase state and occurrence of liquid-liquid phase separation (Marsh et al., 2017). All this information coupled, with the thorough chemical characterisation of the SOA samples described in the previous sections, represents a unique set of experimental observations that will allow a deeper understanding of the complexity behind the dynamics of SOA in the

atmosphere.

To prepare the SOA samples from CFR experiments for the analysis in a CC-EDB, the collected aerosol mass was extracted in a 50:50 water:ethanol mixture (SOA mass fraction of ~ 0.02). The extracts were stored at – 20 °C in a freezer to minimise any evaporative losses. Single charged droplets were generated from the extract solutions using a micro dispenser (Microfab

MJ-ABP-01, initial radius of ~ 18-25 μm) and trapped within ~ 100 ms from generation in the EDB electrodynamic field, up to timescales of days. Once trapped, an individual droplet sits within a nitrogen flow with controlled temperature and RH (gas flow velocity of 3 cm s$^{-1}$, RH range of 0-95 %, T range of 257-313 K). Single particles are also illuminated with light from a 532 nm wavelength laser and the scattered light, centred on an angle of 45°, is collected by means of a CCD camera. The angularly resolved scattering pattern can be used to estimate the evolving size of the droplet by applying the geometric optics

approximation (Glantschnig and Chen, 1981), or on both size and refractive index by fitting the generated phase functions with Mie Theory simulations (using procedure reported by Cotterell et al. (2015)). The experimental setup and operation in this work has been extensively described in previous publications (Davies et al., 2012; Rovelli et al., 2016) and we refer to these for a more detailed description.



## 3. Results

### 3.1 Preliminary SOA characterisation

In total, 38 experiments were performed in the CFR to generate SOA mass from the photo-oxidation of α-pinene, limonene, β-caryophyllene and toluene for offline chemical composition and single particle analysis. The experiments investigated the

effect of: (i) relative humidity, (ii) VOC/NO$_x$ ratios, (iii) combined effect of relative humidity and VOC/NO$_x$ ratios, and (iv) VOC mixing ratios (α-pinene experiments only). The reactor conditions, reactant mixing ratios and the amount of SOA mass collected in each experiment is shown in Table 1. The reactor temperature remained stable throughout all experiments with an average of 24.1 ± 1.0 °C. The average experimental duration was 5 hours and 45 minutes. The total amount of SOA mass collected in each experiment (*i.e.* sum of ELPI + filter) ranged from 42 to 322 mg, excluding the low α-pinene mixing ratio

(range = 5 to 7 mg) and chamber background experiments (see Table 1). The reactor conditions and SOA formation in a typical CFR experiment are shown in Figure 2. The reactor temperature increased by 1.4 °C after turning the UV lamp on, stabilising ~ 50 minutes into the experiment shown in Figure 2. Background SOA mass and number concentrations were 8.1 ± 2.0 µg m$^{-3}$ and 6.9 × 10$^3$ ± 1.7 × 10$^3$ particles cm$^{-3}$, respectively. In contrast to atmospheric simulation chambers where reactants are typically introduced at the beginning of an experiment, the continuous introduction of reactants into the CFR resulted in a

stable formation of SOA mass and number concentrations (Figure 2). Particle number concentrations increased significantly upon UV radiation. The maximum number of particles observed in the typical CFR experiment shown in Figure 2 was 1.7 × 10$^8$ particles cm$^{-3}$, which plateaued ~ 10 minutes into the experiment at 1.3 ± 0.24 × 10$^7$. Following particle nucleation, SOA mass gradually increased plateauing ~ 50 minutes into the experiment at 43.6 ± 1.0 mg m$^{-3}$. The total amount of SOA mass collected in this experiment was 161 mg over 6 hours and 29 minutes.

Visual and physical differences were observed in the SOA samples generated from the experiments shown in Table 1. The SOA mass formed from the photo-oxidation of β-caryophyllene and toluene under replicate experimental conditions (exp. 31 and 36, respectively) are shown in Figure 3. Both experiments investigated SOA formation at 55 % relative humidity with a VOC/NO$_x$ of 13. From Figure 3, it can be observed that the SOA samples display considerable visual differences. The SOA

formed from the photo-oxidation of toluene is yellow/brown in colour displaying strong light absorbing properties in the visible spectrum. Conversely, the SOA formed from the photo-oxidation of β-caryophyllene is translucent, suggesting the SOA has negligible visible light absorbing properties. The SOA mass with the strongest light absorbing properties was formed under the lowest VOC/NO$_x$ ratios (highest NO$_x$ mixing ratios), as observed in Figure S1 and previous studies (Liu et al., 2016; Nakayama et al., 2010; Xie et al., 2017). The viscosity of the SOA samples also appeared to decrease with increasing relative

humidity, an observation which is commonly reported in the literature (Bateman et al., 2015; Kidd et al., 2014; Koop et al., 2011; Mikhailov et al., 2009; Montgomery et al., 2015; Reid et al., 2018; Rothfuss and Petters, 2017a).





The FTIR spectra of the SOA formed from the photo-oxidation of α-pinene, limonene, β-caryophyllene and toluene under replicate experimental conditions (55 % relative humidity with a VOC/$NO_x$ ratio of 3) are shown in Figure 4. The absorbance frequencies of 5 chemical functionalities are highlighted in Figure 4. These absorbance frequencies correspond to an alcoholic hydroxyl (wavenumber 3100 - 3600 $cm^{-1}$), aliphatic (3000-2850 $cm^{-1}$), carbonyl (1750-1680 $cm^{-1}$), nitrate (~ 1630 $cm^{-1}$) and

an aromatic nitro group (1535-1525 $cm^{-1}$) (see Cao et al. 2018 and references therein for wavenumber assignments). Several expected compositional differences were observed in the SOA samples. For example, β-caryophyllene SOA displayed the highest intensity aliphatic functionality. β-caryophyllene has a molecular formula of $C_{15}H_{22}$ with a molecular weight of 204.36 g $mol^{-1}$. α-pinene ($C_{10}H_{16}$), limonene ($C_{10}H_{16}$) and toluene ($C_7H_8$) all contain considerably fewer carbon and hydrogen atoms in their molecular formula. β-caryophyllene was thus the largest chemical structure investigated, accounting for the high

intensity aliphatic peak observed. Organic nitrate functionalities are formed in the presence of NO *via* the reaction $RO_2 + NO$ (*e.g.* (Kroll and Seinfeld, 2008)). All the SOA samples shown in Figure 4 were formed in presence of NO and contained nitrate functionalities. Moreover, the aromatic nitro functionality (formed in the presence of $NO_2$) was only observed in the SOA generated from the photo-oxidation of toluene (an aromatic VOC).

### 3.2 CFR capabilities

Typical chamber experiments generate < 100 µg of SOA mass per experiment (*e.g.* (Nozière et al., 2015; Xie et al., 2017)). The design and operation of the CFR allowed considerable quantities of SOA mass to be collected per experiment (> $10^2$ mg). This considerable quantity of SOA mass allowed us to use compositional and single particle techniques which could not have otherwise been used (techniques requiring ~ 1 to 50 mg of SOA mass per analysis). Here, we take advantage of the opportunity provided by the CFR to compare some of the critical chemical and physical properties of SOA from different precursors,

previously not possible. The following discussion is limited to the characterisation and capabilities of the CFR. A separate publication will discuss the composition and chemical-physical properties findings of this work.

### 3.2.1 Elemental composition

The elemental composition of the SOA samples was determined using a CHNS elemental analyser. CHNS analysis offers high accuracy and precision but is rarely used within aerosol science due to the large amount of organic aerosol mass required per

analysis (1 - 5 mg). Elemental composition is usually determined using aerosol mass spectrometry (AMS) or electrospray ionisation ultra-high resolution mass spectrometry (Liu et. al 2018; Tasoglou and Pandis 2015; Tuet. al 2017; Zhao et. al 2015; Kroll et. al 2011 and references therein). Whilst both mass spectrometric techniques are invaluable within aerosol science, both methods suffer from inaccuracies, either through the use of a selective ionisation source (*i.e.* ESI) or assumptive corrections in AMS data processing (Canagaratna et al., 2015). The H/C and O/C ratios and the average carbon oxidation state

($\overline{OS}_c = 2 \times O/C - H/C$, see Kroll et. al 2011 for further information) were calculated from the CHNS data for each SOA sample. The O/C and H/C ratios ranged from 0.41 – 0.45 and 1.57 – 1.67 for α-pinene SOA, 0.45 – 0.49 and 1.59 – 1.71 for limonene SOA, 0.22 – 0.36 and 1.60 – 1.67 for β-caryophyllene SOA and 0.78 – 0.84 and 1.35 – 1.45 for toluene SOA, respectively.





The elemental composition of two SOA samples formed under replicate conditions (exp. 22 and 23, see Table 1) displayed excellent agreement, with the O/C and H/C ratios observed to be within error of the analytical instrumentation (instrument accuracy ± 0.15% for a reference standard as quoted by the manufacturer (EAI, 2018)).

A Van Krevelen diagram showing the H/C *vs.* O/C ratios of the SOA samples generated in this study with a comparison to literature values, is shown in Figure 5. The high VOC mixing ratios used in this study will result in higher volatility oxidation products partitioning into the particulate phase than observed at lower (ambient) mixing ratios, likely affecting the observed chemical composition (Donahue et al., 2006; Pankow, 1994a, b). However, from Figure 5 it can be observed that the H/C and O/C ratios of the SOA samples display very good agreement with the literature values, suggesting that for the experimental

conditions investigated, the bulk elemental composition is largely unaffected by the use of high VOC mixing ratios. Laboratory and ambient OA has been found to follow a general trajectory in the Van Krevelen diagram (Chen et al., 2015; Heald et al., 2010; Ng et al., 2011a). An approximate -1 slope was first proposed by Heald et. al (2010) and was later re-evaluated to an approximate slope of -0.5 (Chen et al., 2015; Ng et al., 2011b). The SOA samples generated in this study were consistent with the characteristic Van Krevelen diagram trajectory, with a slope of -0.41 observed for all SOA precursors investigated.

The ability to use a highly accurate CHNS elemental analyser to determine the elemental composition of the SOA samples, allowed us to evaluate the accuracy of SOA elemental compositions obtained using UHRMS. The O/C and H/C ratios were calculated from the UHRMS using the assigned molecular formulae in each SOA sample. The total number of compounds (sum of positive and negative ionisation mode) identified in each SOA sample ranged from 100 to 910, see Table S1. Peak

area weighted O/C and H/C ratios were calculated using the equation shown in Bateman et al. (2009), substituting peak intensity for peak area. In all of the experiments performed, the O/C ratios obtained for each SOA sample using UHRMS were lower than the O/C ratios obtained from the CHNS analyser (the non-selective and more accurate technique), see Figure S2. The SOA ΔO/C ratios varied for each SOA precursor, with an average ΔO/C of 0.08 ± 0.01, 0.13 ± 0.01, 0.06 ± 0.03 and 0.36 ± 0.08 for α-pinene, limonene, β-caryophyllene and toluene SOA, respectively. For toluene SOA, this means that the average

O/C ratio would have been under reported by 45 % if obtained from UHRMS (using the data processing methods described, see section 2.3.3) rather than the CHNS elemental analyser. A linear relationship ($R^2 = 0.9222$) was observed for the O/C ratios obtained from UHRMS *vs.* CHNS elemental analyser for the β-caryophyllene SOA samples, suggesting with further work, a possible correction factor could be applied to UHRMS generated SOA O/C ratios to correct for the inaccuracy in the use of a selective ionisation source. Further work is required to investigate the accuracy of SOA O/C ratios obtained from UHRMS,

including the effect of different SOA precursors, introduction techniques (*i.e.* liquid chromatography or direct infusion) and data processing methods. This investigation demonstrates the capabilities and use of the CFR, allowing sufficient quantities of SOA mass to be generated in order to evaluate the accuracy of existing techniques.





### 3.2.2 NMR spectroscopy

NMR spectroscopy can provide detailed structural information on the carbon, hydrogen and nitrogen nuclei bonds present in complex mixtures. This technique is complementary to detailed chemical speciation, providing molecular level insight into the bulk chemical composition of OA (Chalbot and Kavouras, 2014; Duarte and Duarte, 2011, 2015; Simpson et al., 2012). Proton NMR spectroscopy ($^1$H NMR) is the most commonly used NMR technique for the analysis of OA, although numerous alternative nuclei and two-dimensional spin-spin coupling methods exist (see Duarte and Duarte (2015) for further information). The complexity of OA can reduce the amount of chemical information that can be obtained from $^1$H NMR analysis. Ambient OA contains thousands of compounds of differing chemical functionalities, which combined with the relatively small dispersion of proton chemical shifts (0 to ~ 10 ppm), often results in a complex unresolved spectrum with few abundant peaks (Duarte and Duarte, 2011). The binning of spectral regions which correspond to certain functionalities (*e.g.* aliphatic protons, aromatic protons *etc.*) *via* offline data processing, can aid in the compositional interpretation of bulk OA using $^1$H NMR analysis (Decesari et al., 2000; Decesari et al., 2001) and has been successfully used in several studies (see Chalbot and Kavouras (2014) and references therein). Two dimensional NMR spectroscopy however, can provide increased resolution and additional compositional information in comparison to $^1$H NMR spectroscopy, although is rarely used within aerosol science due to the large amount of mass required for analysis (> 1 mg) (Duarte and Duarte, 2015; Simpson et al., 2012). Here we show, as an example, how the large quantity of SOA mass collected from the CFR can be used to gain greater compositional insights into SOA using two-dimensional NMR spectroscopy.

The SOA samples were analysed using $^1$H and two-dimensional $^1$H-$^{13}$C heteronuclear single-quantum correlation (HSQC) NMR spectroscopy. HSQC detects the carbon-hydrogen couplings of each bond in a molecular substructure, providing the proton and carbon shifts for both atoms. The $^1$H and $^1$H-$^{13}$C HSQC NMR spectra of β-caryophyllene and toluene SOA formed at 55 % relative humidity and VOC/NO$_x$ ratio of 3 (exp. 31 and 36 Table 1, respectively) are shown in Figure 6. The $^1$H NMR spectral regions as defined in Decesari et. al (2000) and (2001) are shown in Figure 6A and C. The $^1$H NMR spectra of β-caryophyllene and toluene SOA are vastly different. β-caryophyllene SOA displays an abundance of aliphatic proton groups ($\delta^1$H 0.7 to 3.2 ppm) with hardly any peaks observed for protons bonded to oxygenated saturated aliphatic carbon atoms ($\delta^1$H 3.4 to 4.1 ppm and $\delta^1$H 5.0 to 5.6), confirming the SOA sample is not very oxidised (CHNS data, O/C ratio = 0.31). The $^1$H NMR spectra of toluene SOA displays an abundance of protons bonded to an adjacent aliphatic carbon-carbon double bond ($\delta^1$H 1.9 to 3.2 ppm) and (dissimilar to β-caryophyllene SOA) oxygenated saturated aliphatic carbon atoms ($\delta^1$H 3.4 to 4.1 ppm), suggesting the sample contains ring opened species and is highly oxidised (CHNS data, O/C ratio = 0.84). In contrast to the $^1$H NMR spectrum of β-caryophyllene SOA, toluene SOA displays protons bonded to an aromatic carbon atom (ring retaining species) and an unresolved low intensity peak for protons bonded to aliphatic methyl, methylene and/or methyne groups ($\delta^1$H 0.7 to 1.9 ppm).



The $^1$H-$^{13}$C HSQC NMR spectra displays considerably more compositional information than observed in the $^1$H NMR spectra (see Figure 6). The spectral regions shown in the $^1$H-$^{13}$C HSQC spectra have been adapted from Chen et. al (2016) using a structural carbon-hydrogen coupling predictive tool for known and structurally similar β-caryophyllene and toluene oxidation products (see Section 2.3.2 for further information). The abundant aliphatic protons (δ$^1$H 0.7 to 3.2 ppm) observed in the $^1$H NMR spectra of β-caryophyllene SOA (Figure 6A), displays over 40 carbon-hydrogen coupling signals in the HSQC spectra (Figure 6B). The compositional benefits of $^1$H-$^{13}$C HSQC analysis can be observed in region I of Figure 6A and B. The most abundant peak in the $^1$H NMR spectrum of β-caryophyllene SOA is from protons bonded to a methyl group (δ$^1$H 1.1 ppm). In the $^1$H-$^{13}$C HSQC spectrum, it can be observed that the methyl group at δ$^1$H 1.1 ppm consists of two carbon-hydrogen coupling signals (*i.e.* the same proton chemical shift but different carbon chemical shifts). The two carbon-hydrogen coupling signals correspond to a methyl group bonded to an adjacent tertiary carbon atom at δ$^{13}$C 21.2 ppm and an adjacent cyclic carbon atom at δ$^{13}$ 29.9 ppm, both which are bonded to a neighbouring methyl group (resembling the carbon-backbone structure of β-caryophyllene). This compositional information cannot be observed in the $^1$H NMR spectra. Similarly, additional compositional information can be observed in the $^1$H-$^{13}$C HSQC spectrum of toluene SOA. Region II in the $^1$H NMR spectra displays an abundant aliphatic proton peak at δ$^1$H 2.0 ppm. In the HSQC spectrum, two defined carbon-hydrogen coupling signals can be observed. Based on predicted carbon-hydrogen couplings, these two signals likely represent the carbon-hydrogen bonds in the methyl group attached to an aromatic ring with (δ$^{13}$C 19.1) and without (δ$^{13}$C 12.5 ppm) an electronegative aromatic substituent, such as a hydroxyl group, in the *meta* position to the methyl group. In addition, region III likely displays aromatic carbon-hydrogen bonds which are in the *meta* position to a nitro aromatic substitution; the only aromatic substituent which would result in an adjacent carbon-hydrogen chemical shift of δ$^{13}$C ~125 ppm and δ$^1$H ~ 8 ppm, respectively. It is imperative to note that predictive carbon-hydrogen coupling tools should only be used as a guide and authentic standards (where possible) should be used to confirm predicted carbon-hydrogen coupling shifts. Nevertheless, the above discussion demonstrates the additional compositional information which can be obtained using two-dimensional NMR spectroscopy. Few studies have used two-dimensional NMR spectroscopy for the compositional analysis of bulk OA. This technique in combination with the use of a spectral library, has the capability to aid in the interpretation of detailed chemical speciation and provide new insights into the bulk chemical composition of SOA.

### 3.2.3 Generating SOA standards

ESI-UHRMS is capable of providing molecular and structural speciation of individual compounds in complex mixtures and is therefore widely used within aerosol science to determine the detailed chemical speciation of OA (Laskin et al., 2012; Nizkorodov et al., 2011; Pratt and Prather, 2012). The lack of commercially available standards however, often results in the use of surrogate standards (*i.e.* structurally/compositionally similar species) for quantification. The use of surrogate standards can have a considerable effect on the reported concentrations when using a selective ionisation source, such as ESI. The molecular size, volatility, basicity and polarity of a compound can all effect the ionisation response (Kiontke et al., 2016; Oss





et al., 2010). Here, we show how the large quantity of SOA mass collected from the CFR can be used to generate SOA standards, reducing the need for additional commercially available standards or chemical synthesis.

Several experiments, in addition to those shown in Table 1, were performed to generate additional SOA mass from the photo-
oxidation of α-pinene. This additional SOA mass was used to generate standards to quantify components in the SOA mass formed in the α-pinene experiments. A similar methodology has previously been used in our group, generating standards from SOA mass formed in a micro-reactor (see Finessi et al. (2014)). In total, 17 compounds were targeted in the generated SOA mass. HPLC-ITMS coupled to an automated fraction collector was used to isolate and collect the targeted compounds based on their retention times. The molecular identification of each standard was determined using a combination of the molecular
information and fragmentation patterns provided by the UPLC-UHRMS[2] and the proton chemical shifts obtained from [1]H NMR spectroscopy. [1]H NMR spectroscopy was used to determine the amount of mass collected for each targeted compound *via* the integration of the peak integrals of a known proton peak (*e.g.* a methyl group) and the internal standard (*i.e.* TSP). Once the concentration of each fraction had been determined, the standard was used for quantification. Major α-pinene oxidation products often contain two characteristic methyl groups attached to a cyclic ring (*e.g.* pinonic acid, pinonaldhyde, pinic acid,
10-hydroxypinonic acid, among others). The characteristic cyclic methyl groups can easily be identified in the [1]H NMR spectrum (Finessi et al., 2014). Several of the targeted compounds contained the two characteristic cyclic methyl groups, allowing the concentration of the standard to be determined (*via* the integration of the methyl protons) even if the entire chemical structure was unclear. Of the 17 targeted compounds, 10 were deemed suitable for use as standards. The other 7 compounds were excluded from further analysis due to an insufficient amount of collected mass and/or complex spectrum,
where the cyclic methyl groups could not be identified preventing the concentration of the standard from being determined. The molecular formulae, retention times and collection times of the 10 standards are shown in Table S2.

A comparison of the chromatographic peaks obtained for both the generated and authentic standard of pinic acid at a concentration of 1 ppm is shown in Figure S3. From Figure S3, it can be observed that both chromatographic peaks display
relatively good agreement, with similar peak shapes and ion intensities observed (10 % difference in peak area). In addition to the 10 targeted compounds, 4 α-pinene standards which were generated in Finessi et al. (2014) and 2 commercially available compounds (*i.e.* pinic acid and cis-pinonic acid) were used to identify and quantify components in the α-pinene SOA samples using UPLC-UHRMS. An identification was confirmed if a compound in the SOA sample displayed the same molecular formula (< 2 ppm error), retention time (± 30 seconds) and fragmentation patterns as the standard. Calibrations were performed
for all of the compounds shown in Table S2, with the exception of compound 5, where the authentic pinic acid standard was used instead. Calibrations ranged from 0.001 to 15 ppm with a minimum of 4 concentrations and 3 replicate measurements per concentration. The total amount of SOA mass quantified in each experiment is shown in Figure 7. The standards represented up to 35.8 ± 1.6 % of the total α-pinene SOA mass. The error represents the propagated uncertainty in the slope of each calibration graph used for quantification. The quantified α-pinene SOA mass varied considerably depending on the





experimental conditions. The average amount of α-pinene SOA mass quantified in the no $NO_x$ and $NO_x$ experiments was 6.9 and 33.2 %, respectively. The targeted standards were selected in α-pinene SOA formed under $NO_x$ conditions, likely accounting for the larger amount of SOA mass quantified in the $NO_x$ experiments. Molecular speciation techniques typically quantify < 25 % of OA mass (Hallquist et al., 2009; Nozière et al., 2015). Using the techniques described, considerable
improvements can be made in the total amount of SOA mass quantified.

### 3.2.4 Chemical and physical properties of SOA single particles

SOA can exist in highly viscous semi-solid and solid states, depending on chemical composition and the conditions of the surrounding gas phase (T and RH); the phase state of SOA particles affects the diffusion rates of molecules within the aerosol phase, affecting physicochemical processes such as the partitioning of organic species between the condensed and gas phases,
heterogeneous chemical reactions and ice nucleation (Reid et al., 2018). As indicated by Shrivastava et al. (2017), there is the need of a systematic investigation of the viscosity of SOA and its effects on the physicochemical properties of SOA particles as a function of T and RH and for SOA formed from different precursors at variable $NO_x$ and RH conditions. The CFR experiments presented in this work, coupled with the thorough information on the SOA chemical composition (sections 3.2.1-3.2.3) and the single particle experiments described below, represent a unique opportunity for such a systematic investigation.

As an example of the capabilities of the single particle EDB approach to elucidate the physiochemical properties of the SOA samples, we report in Figure 8 a comparison of the volatility of β-caryophyllene (exp. 28, see Table 1) and toluene SOA (exp. 35) formed in the CFR at 55% RH with no $NO_x$. Once a diluted SOA droplet is initially trapped (typical initial SOA extract mass fraction of ~ 0.02), a steep decrease in size is observed with both water and ethanol evaporating to reach equilibrium
with the gas phase composition (timescale of less than 10 seconds at 293 K). No ethanol is present in the gas phase and therefore, it evaporates completely from the droplet; by contrast, the water content in the condensed phase equilibrates such that the water activity in solution is equal to that of the gas phase. After this rapid evaporation phase, the radius of SOA trapped single droplets is generally observed to decrease slowly over time, with the semi-volatile organic components within the droplets partitioning to the gas phase together with the solvating water. In Figure 8, the loss of these organic components (and
solvating water) is represented as a volume fraction remaining (VFR) compared to the "initial" droplet volume once the initial loss and equilibration of water content has occurred, identified as the point where the rapid water/ethanol evaporation is concluded (typical initial reference radius of ~ 5-8 μm). To report the VFR values in Figure 8, the measured radii were separated into bins, each corresponding to a $\log_{10}$ (time) = 0.1 interval and then averaged.

Figure 8A compares VFR data for two β-caryophyllene SOA particles evaporating into high RH and dry conditions. In both cases, the VFR decreases with time indicating a mass loss from the particle, due to the volatilisation of organic species, and the two curves present a similar trend. The activity of the organic species at 0 % RH is expected to be higher than at 85 % RH, as a consequence of the absence of condensed phase water and a higher concentration of the organic components. Although a


faster evaporation rate would be expected under dry conditions compared to wet conditions. As a result, the two time-dependencies in the VFR datasets in Figure 8A show a very similar trend. This can be likely explained by a kinetic limitation on the diffusion of the organic components within the β-caryophyllene SOA particle at 0 % RH due to high viscosity. Indeed, Li et al. (2015) measured a transition of β-caryophyllene SOA particles from liquid to non-liquid at a RH above 90 % from

particle bounce experiments, supporting the hypothesis that a high viscosity is restricting the rate of volatilisation from the β-caryophyllene SOA particle at low RH. The secondary *x*-axis in Figure 8 represents an estimate of the evolving effective saturation concentration ($C^*$) for a 7 µm droplet, calculated according to Donahue et al. (2006) and assuming a diffusion coefficient of $10^{-6}$ m$^2$ s$^{-1}$, a molecular mass of 200 g mol$^{-1}$ and a density of 1.4 g m$^{-3}$. This calculation provides an estimate of the lifetime of each of the indicated $C^*$ bins in the case of no kinetic limitations to the evaporation of the organic molecules.

For example, organic molecules in the $10^2$ µg m$^{-3}$ are expected to have a lifetime in the condensed phase < 1000 s and at the end of the experimental timescale (~ $10^5$ s), only molecules with $C^*$ lower than $10^{-1}$ µg m$^{-3}$ are still present in the evaporating droplet.

When compared to β-caryophyllene SOA, toluene SOA particles present a very different VFR profile (Figure 8B): significant

evaporation of the trapped particle is observed at high RH, but the evaporative loss of semi-volatiles appears completely inhibited at 0 % RH after ~$10^2$ s, with the particle size achieving a constant value. Similarly, but more markedly when compared to the β-caryophyllene SOA case, this complete inhibition of the volatilisation of organic components from the condensed phase is caused by the high viscosity of the toluene SOA particle at 0% RH. Song et al. (2016) inferred a lower limit of viscosity for toluene SOA below 17 % RH of ~ $5 \cdot 10^8$ Pa s; the observation of strong kinetic limitation to evaporation shown

in Figure 8B is consistent with such high viscosity. The difference in VFR after $10^5$ s between particles held at wet and dry conditions is significant (~ 0.35) and it is a clear indication that the size of toluene SOA particles strongly depends on their phase state (liquid *vs*. semi-solid). In a future paper, we will provide a comprehensive analysis of all volatilisation measurements, reporting the volatility and viscosity distribution that characterise the various SOA samples in this work by using the KM-GAP model (Yli-Juuti et al., 2017) to analyse the experimental data.

**3.3 CFR limitations**

A comparison of the bulk chemical functionalities observed in two SOA samples formed from the photo-oxidation of α-pinene, with a VOC mixing ratio of 18.5 ppmv (exp. 10) and 2.1 ppmv (exp. 14) are shown in Figure S4. The bulk SOA chemical functionalities were determined using ATR-FTIR spectroscopy. Both experiments were performed at 55 % relative humidity with a VOC/NO$_x$ ratio of 3. The peak heights of the individual chemical functionalities were normalised to the total peak height

of all speciated chemical functionalities in each SOA sample, allowing a direct comparison between samples. It is worth noting that a second low α-pinene mixing ratio experiment was performed (exp. 7, see Table 1). However, this SOA sample displayed poor spectral absorption in all three replicate measurements, the reason for which is unclear. Subsequently, this sample has been excluded from the following discussion. From Figure S4, it can be observed that the aromatic nitro, nitrate and aliphatic



functionalities in both SOA samples display relatively good agreement. However, alcohol and carbonyl functionalities display some disagreement. The SOA formed from the low α-pinene mixing ratio experiment (2.1 ppmv) contained increased alcohol functionality (~ 12%) and decreased carbonyl functionality (~ 11%), in comparison to the SOA formed from the high α-pinene mixing ratio experiment (18.5 ppmv). This discrepancy is likely due to the partitioning of higher volatility species into the

particulate phase with the use of higher VOC mixing ratios (Donahue et al., 2006; Pankow, 1994a, b), effecting the observed chemical functionality. The bulk elemental composition however, did not appear to be largely affected by the use of high VOC mixing ratios (see section 3.2.1). Nevertheless, future studies using lower mixing ratios and extended sampling times could be used to overcome this.

## 4. Conclusion

This study describes the design and operation of a newly built 0.3 m$^3$ CFR which can be used as a tool to gain greater insights into the composition and physical state of SOA. The CFR was used to generate SOA mass from the photo-oxidation of α-pinene, limonene, β-caryophyllene and toluene under different experimental conditions. The design and operation of the CFR allowed > 10$^2$ mg of SOA mass to be collected per experiment. The considerable quantities of SOA mass collected in each experiment, allowed the use of highly accurate compositional and single particle analysis techniques which are not usually

accessible, due to the large amount of OA mass required for analysis. Four techniques were presented (as examples) to demonstrate the additional compositional and physical state information which can be obtained using the methods outlined in this manuscript. The four techniques included, (i) the use of a highly accurate CHNS elemental analyser to determine the elemental composition of the generated SOA samples and the ability to evaluate the accuracy of reported elemental compositions using a commonly used technique (UHRMS), (ii) the additional compositional information which can be

obtained using two-dimensional NMR spectroscopy, (iii) the generation of SOA standards, overcoming the analytical challenges associated with the lack of commercially available standards and (iv) the first use of an electrodynamic balance to assess the influence of the temperature and phase state of the SOA on the volatilisation kinetics of semi-volatile components from a sample particle. High VOC mixing ratios (ppmv levels) were used in this study to generate sufficient quantities of SOA mass for offline analysis. The investigation of two replicate experiments using different α-pinene mixing ratios (18.5 and 2.1

ppmv) did display a slight discrepancy (~ 11% difference) in bulk SOA alcohol and carbonyl functionalities, possibly a result of the high VOC mixing ratios used. However, aliphatic, nitrate and aromatic nitro functionalities all displayed relatively good agreement. It is important to note, that the objective of this study was not to mimic atmospheric conditions, but to provide a tool which allowed the use of highly accurate techniques to gain greater insights into the chemical and physical properties of SOA. Nevertheless, the elemental composition of the generated SOA displayed very good agreement with literature values,

suggesting for the experimental conditions investigated, the bulk elemental composition is largely unaffected by the use of high VOC mixing ratios. The SOA generated from two replicate experiments displayed excellent agreement, with measured O/C and H/C ratios within error of the analytical instrumentation. Using the methods described, we were able to quantify up



to 36 % of α-pinene SOA which is a considerable improvement from most previous studies. The CFR costs ~ £8000 to build including the reactor housing. A considerable proportion of this cost is attributed to use of several mass flow controllers at ~ £6500. The mass flow controllers can be substituted for cheaper alternatives (*e.g.* ball-flow meters) which will significantly reduce the cost. However, due to the reduced accuracy in the flow rates of these alternatives methods, the CFR will need to be

operated at low flow rates (less than ~ 4 Lpm$^{-1}$) and the reactor volume closely monitored. The CFR is incredibly versatile. Multiple instruments can be connected to the reactor and easily interchanged. Different oxidants and/or scavengers can be introduced into the reactor and mixed VOC experiments can be performed (*i.e.* introduction of two VOCs). The CFR can also be designed to be more sophisticated with a simple addition of a software program (using DAQ factory, or similar) for the automated control of the mass flow controllers. This work demonstrates how the unconventional use of a newly built CFR can

used to gain considerable insights into the chemical and physical properties of SOA, providing a greater understanding of the relationship between SOA formation conditions, chemical composition and physicochemical properties.

## 5. Author contribution

K. Pereira designed, built and operated the CFR, designed the experiments, collected and distributed the SOA samples, performed all compositional analysis and associated data interpretation and wrote the manuscript. A. Mayhew aided K. Pereira

in the collection, evaporation and $^1$H NMR preparation of the α-pinene fractions. J. Hamilton was responsible for the conceptualization and funding acquisition of the York component of the project and supervised K. Pereira's work. G. Rovelli was responsible for the single particle analysis and data interpretation of the SOA samples at the University of Bristol. Y. Song assisted G. Rovelli with the single particle analysis and data interpretation. J. Reid was responsible for the conceptualization and funding acquisition of the Bristol component of the project, co-ordinated the research activity planning and execution of

the whole project (lead PI) and supervised the work performed by G. Rovelli and Y. Song. All authors contributed to the manuscript.

## 6. Data availability

The unprocessed ATR-FTIR and $^1$H and $^1$H-$^{13}$C HSQC NMR spectral data shown in Figures 4 and 6, and the tabulated data used to plot Figures 5 and 7 have been provided in a data depository, see the University of York research database PURE

(DOI: provided upon acceptance). The unprocessed EDB data shown in Figure 8 has been provided in the University of Bristol data depository (DOI: provided upon acceptance).

## 7. Competing interests

The authors declare that they have no conflict of interest.



## 8. Acknowledgements

This work was supported by the Natural Environment Research Council NE/M002411/1. The $NO_x$ and $O_3$ analysers used in this work were provided by Dr Katie Read, University of York, through the Atmospheric Measurement Facility (AMF); a facility housed at the Wolfson Atmospheric Chemistry Laboratories (WACL) funded through the National Centre for Atmospheric Science (NCAS). The Orbitrap was funded by a Natural Environment Research Council strategic capital grant CC090.

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



**Table 1** – Experimental descriptions, reactor operating conditions, VOC and oxidant mixing ratios and amount of SOA mass collected

| Exp. | Exp. Description | RH (%) | NOx | Actual Conditions* | | Exp. Conditions | | | SOA Mass Collected | |
|---|---|---|---|---|---|---|---|---|---|---|
| | | | | RH (%) | Temperature (°C) | VOC (ppmv)† | VOC:NOx | Exp. Duration (HH:MM) | ELPI (mg) | Filter (mg) |
| 1 | Chamber background | | | < LOD | 25.8 ± 0.8 | - | - | 04:34 | -[a] | 0.14 |
| 2 | α-pinene | 0 | - | < LOD | 26.0 ± 0.4 | 26.3 | - | 05:15 | 46.03 | 31.08 |
| 3 | α-pinene | 20 | - | 21.3 ± 0.5 | 26.0 ± 0.4 | 26.3 | - | 04:18 | 82.02 | 61.69 |
| 4 | α-pinene | 40 | - | 38.3 ± 1.4 | 25.8 ± 0.5 | 26.2 | - | 05:00 | 103.73 | 73.53 |
| 5 | α-pinene | 55 | - | 51.0 ± 1.5 | 23.9 ± 0.3 | 26.2 | - | 05:17 | 130.21 | 88.67 |
| 6 | Chamber background | | | < LOD | 24.1 ± 0.2 | - | - | 02:31 | -[a] | 0.14 |
| 7 | α-pinene, low mixing ratio | 55 | Low | 58.7 ± 0.6 | 23.2 ± 0.5 | 2.1, 6.2[f] | 13.0 | 04:25 | -[a] | 5.54 |
| 8 | α-pinene | 55 | Low | 52.5 ± 1.6 | 24.1 ± 0.4 | 18.5 | 13.0 | 05:04 | 122.15 | 102.83 |
| 9 | α-pinene | 55 | Medium | 51.8 ± 1.9 | 25.4 ± 0.4 | 18.5 | 7.6 | 05:03 | 113.91 | 76.97 |
| 10 | α-pinene | 55 | High | 50.3 ± 1.3 | 26.2 ± 0.4 | 18.5 | 2.8 | 05:20 | 122.01 | 85.91 |
| 11 | α-pinene | 20 | Low | 17.6 ± 0.9 | 25.9 ± 0.4 | 18.5 | 13.0 | 05:08 | 88.56 | 65.17 |
| 12 | α-pinene | 20 | Medium | 18.0 ± 0.6 | 26.2 ± 0.6 | 18.5 | 7.6 | 06:37 | 120.91 | 87.83 |
| 13 | α-pinene | 20 | High | 19.2 ± 1.0 | 24.4 ± 0.3 | 18.5 | 2.8 | 04:08 | 75.95 | 58.53 |
| 14 | α-pinene, low mixing ratio | 55 | High | 48.3 ± 1.4 | 23.6 ± 0.3 | 2.1 | 2.7 | 02:19 | -[a] | 6.93 |
| 15 | Chamber background | | | 49.1 ± 1.5 | 22.9 ± 0.2 | - | - | 06:37 | -[a] | 1.67 |
| 16 | Limonene | 0 | - | < LOD | 24.2 ± 0.3 | 18.4 | - | 05:07 | -[a] | 52.75 |
| 17 | Limonene, repeat | 0 | - | < LOD | 24.5 ± 0.3 | 23.1 | - | 06:10 | 47.59 | 84.21 |
| 18 | Limonene | 20 | - | 16.9 ± 0.9 | 23.9 ± 0.3 | 23.1 | - | 05:50 | 74.69 | 69.05 |
| 19 | Limonene | 55 | - | 49.8 ± 1.4 | 23.6 ±0.3 | 23.1 | - | 06:00 | 82.78 | 73.34 |
| 20 | Limonene | 20 | Low | 17.5 ± 0.9 | 24.5 ± 0.3 | 23.1 | 13.0 | 06:29 | 77.06 | 84.19 |
| 21 | Limonene | 20 | High | 18.5 ± 1.2 | 24.4 ± 0.2 | 23.1 | 2.8 | 05:20 | 56.84 | 84.21 |
| 22 | Limonene | 55 | Low | 49.9 ± 1.3 | 23.3 ± 0.4 | 23.1 | 13.0 | 06:48 | 97.50 | 77.41 |
| 23 | Limonene, repeat | 55 | Low | 48.7 ± 2.3 | 23.1 ± 0.4 | 23.1 | 13.0 | 06:47 | 107.65 | 85.77 |
| 24 | Limonene | 55 | High | 46.2 ± 1.7 | 23.2 ± 0.5 | 23.1 | 2.8 | 06:47 | 96.04 | 81.49 |
| 25 | Chamber background | | | 50.2 ± 1.0 | 23.1 ± 0.4 | - | - | 03:35 | -[a] | 0.11 |
| 26 | β-caryophyllene | 0 | - | < LOD | 24.3 ± 0.4 | 63.4 | - | 06:05 | 137.44 | 113.52 |
| 27 | β-caryophyllene | 20 | - | 16.0 ± 0.9 | 24.0 ± 0.3 | 63.4 | - | 07:28 | 176.74 | 145.14 |
| 28 | β-caryophyllene | 55 | - | 44.1 ± 0.7 | 23.2 ± 0.4 | 63.4 | - | 06:24 | 93.00 | 66.83 |
| 29 | β-caryophyllene | 20 | Low | 15.7 ± 0.7 | 24.4 ± 0.1 | 45.6 | 13.0 | 06:34 | 141.64 | 99.73 |
| 30 | β-caryophyllene | 20 | High | 17.4 ± 2.1 | 23.9 ± 0.3 | 45.6 | 2.8 | 07:22 | 59.22 | 42.83 |
| 31 | β-caryophyllene | 55 | Low | 43.1 ± 1.9 | 23.5 ± 0.3 | 45.6 | 13.0 | 06:47 | 122.98 | 74.49 |
| 32 | β-caryophyllene | 55 | High | 41.2 ± 1.1 | 23.1 ± 0.2 | 45.6 | 2.8 | 07:26 | 44.56 | 32.51 |
| 33 | Chamber background | | | 41.7 ± 2.2 | 24.6 ± 0.5 | - | - | 06:59 | -[a] | -[e] |
| 34 | Toluene[b, c] | 55 | - | 41.7 ± 2.0 | 24.1 ± 0.5 | 23.1 | - | 05:21 | 21.06 | 20.98 |
| 35 | Toluene[b, d] | 55 | - | 41.4 ± 1.6 | 24.0 ± 0.5 | 23.1 | - | 07:22 | 26.00 | 29.72 |
| 36 | Toluene[b, c] | 55 | Low | 39.5 ± 2.1 | 23.8 ± 0.5 | 23.1 | 13.0 | 05:45 | 29.48 | 29.26 |





| 37 | Toluene[b, d] | 55 | Low | 39.5 ± 0.1 | 24.3 ± 0.4 | 23.1 | 13.0 | 07:19 | 22.15 | 35.44 |
| 38 | Toluene[c] | 55 | High | 43.1 ± 1.4 | 24.3 ± 0.1 | 23.1 | 2.8 | 07:11 | 30.67 | 28.57 |

[*] = Average reactor conditions during the experiment (*i.e.* from UV lights on to off). † = Estimated mixing ratio (see experimental section 2.1 for further information). [a] = Insufficient SOA mass generated for collection using the ELPI. [b] = Experiments performed in sets to collect sufficient SOA mass. [c] = Compositional analysis only. [d] = Single particle analysis only. [e] = Insufficient mass for gravimetric weighing. [f] = VOC and nitric oxide mixing ratios increased during experiment; insufficient SOA mass formed using starting mixing ratios. RH = relative humidity.





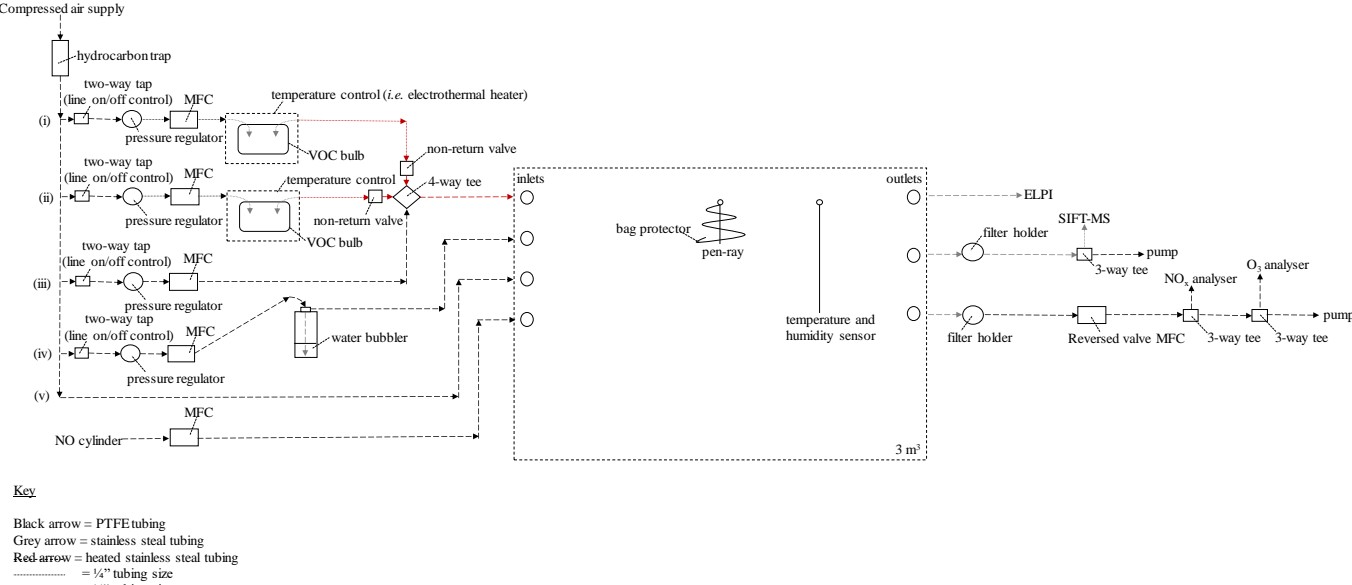

**Figure 1** – Detailed schematic of the continuous flow reactor (CFR). (i) to (iv) refers to the text discussion of each introduction line, see section 2.1.2 introduction system. MFC = mass flow controller. ELPI = electrical low pressure impactor. SIFT-MS = selected ion flow tube mass spectrometer.



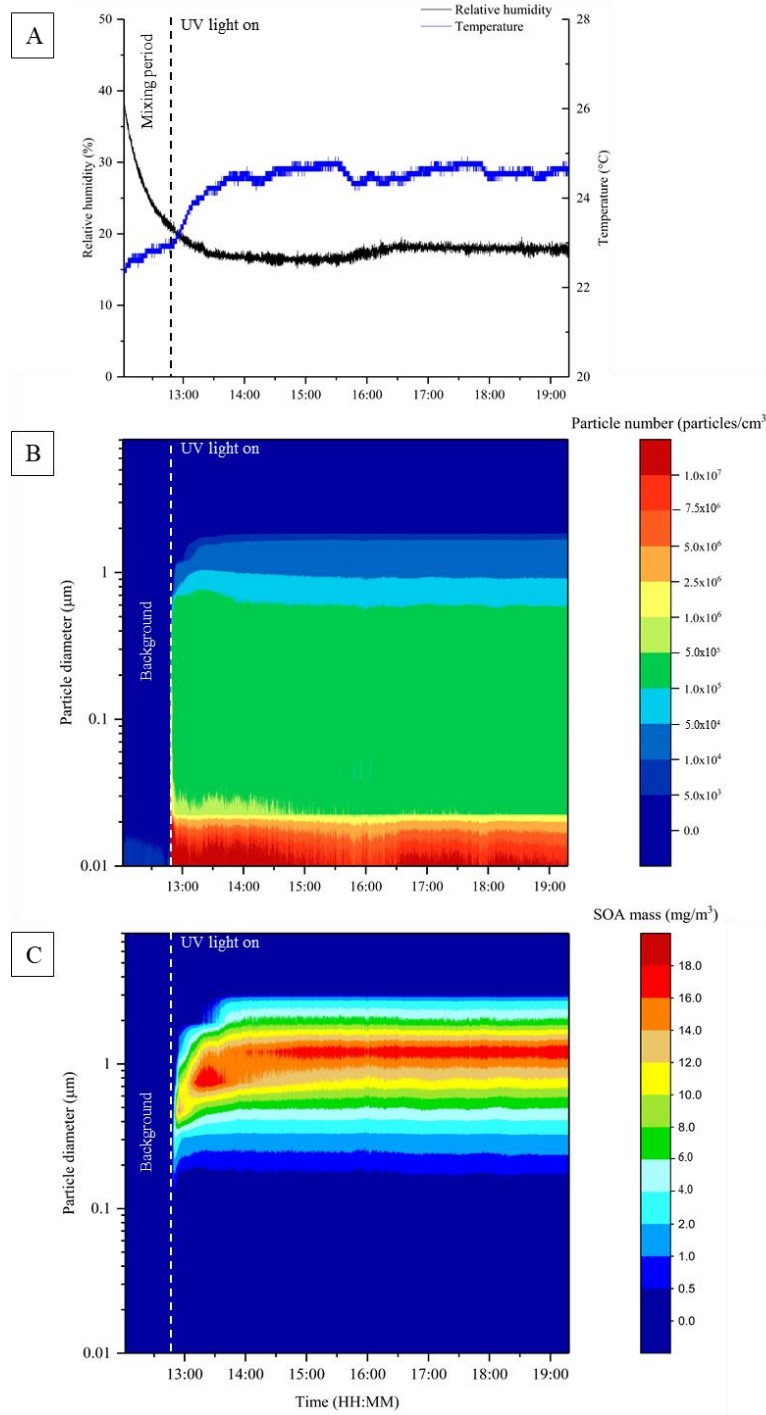

**Figure 2** – Data from a typical CFR experiment, displaying reactor relative humidity and temperature (A), particle diameter and number (B) and particle diameter and mass (C). Measurements are from the photo-oxidation of limonene at 20% relative humidity with a VOC/NO$_x$ ratio of 13 (exp. 20, Table 1).



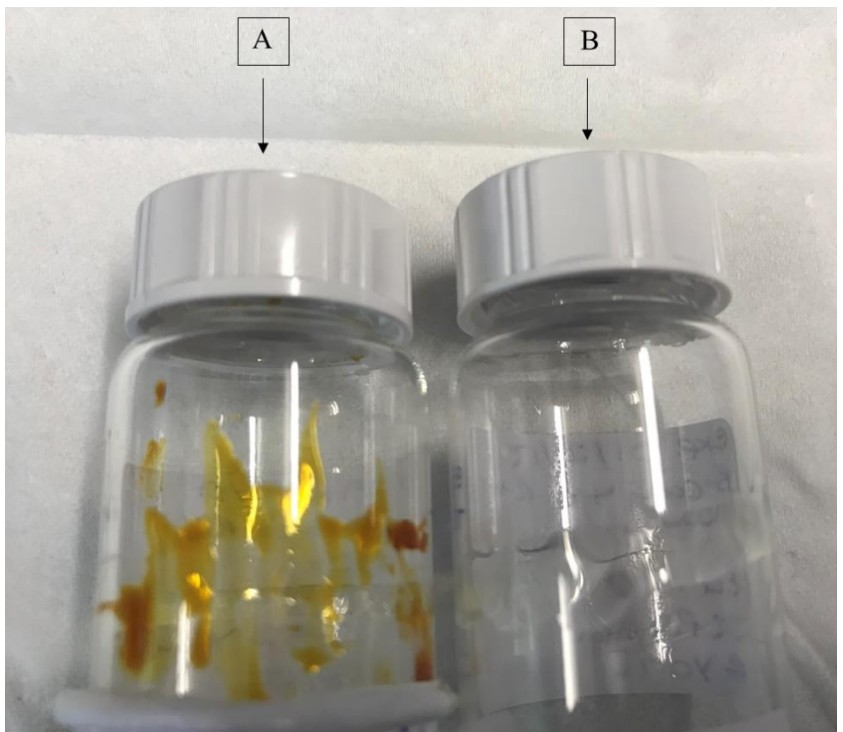

**Figure 3** – Visual differences observed in the light absorbing properties of toluene (A) and β-caryophyllene SOA (B) formed under replicate experimental conditions (55% relative humidity with a VOC/NO$_x$ ratio of 13, exp. 36 and 31 (see Table 1), respectively).



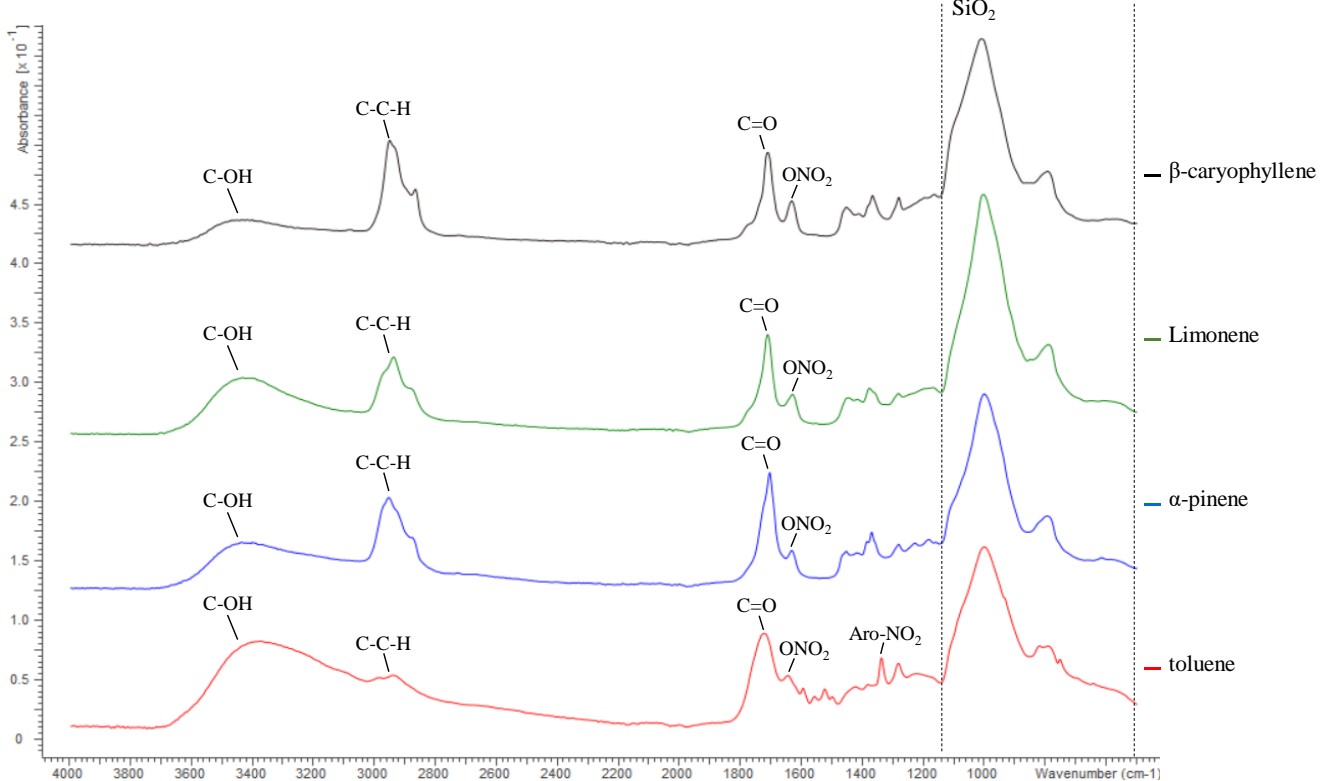

**Figure 4** – ATR-FTIR spectroscopy spectra of β-caryophyllene, limonene, α-pinene and toluene SOA displaying absorption frequencies of organic functional groups. The quartz filter (*i.e.* SiO₂) absorption region is highlighted by a dashed line, see text for further information. Data from experiments 10, 24, 32 and 38, see Table 1. SOA was formed at 55% relative humidity with a VOC/NOₓ ratio of 3 for all SOA samples shown above.





**Figure 5** – Comparison of the elemental hydrogen-to-carbon ratio (H/C, y-axis), oxygen-to-carbon ratio (O/C, x-axis) and average carbon oxidation state ($\overline{OS}_c$, colour scale) of the SOA formed from the photo-oxidation of β-caryophyllene, limonene, α-pinene and toluene in this study (colour unfilled shapes) *vs.* literature values (colour-filled shapes). Letters correspond to the references where the literature values were obtained; b = Bateman et. al (2009), n = Nakao et. al (2013), r = Reinhardt et. al (2007), tu = Tuet et. al (2017), t = Tasoglou and Pandis (2015), h = Huffman et. al (2009), s = Shilling et. al (2009), k = Kim et. al (2014), z = Zhao et. al (2015), c = Chhabra et. al (2010), l = Liu et. al (2018).







**Figure 6** – $^1$H and $^1$H -$^{13}$C HSQC NMR spectra of β-caryophyllene (A and B, respectively) and toluene (C and D, respectively) SOA formed at 55% relative humidity with a VOC/NO$_x$ ratio of 13 (exp. 31 and 36 see Table 1, respectively). The proton spectral regions as defined in Decesari et. al (2000) and (2001) are shown in the $^1$H NMR spectra as dashed black lines. The spectral regions shown in the $^1$H -$^{13}$C HSQC NMR spectra have been adapted from Chen et. al (2016), see text for further information.



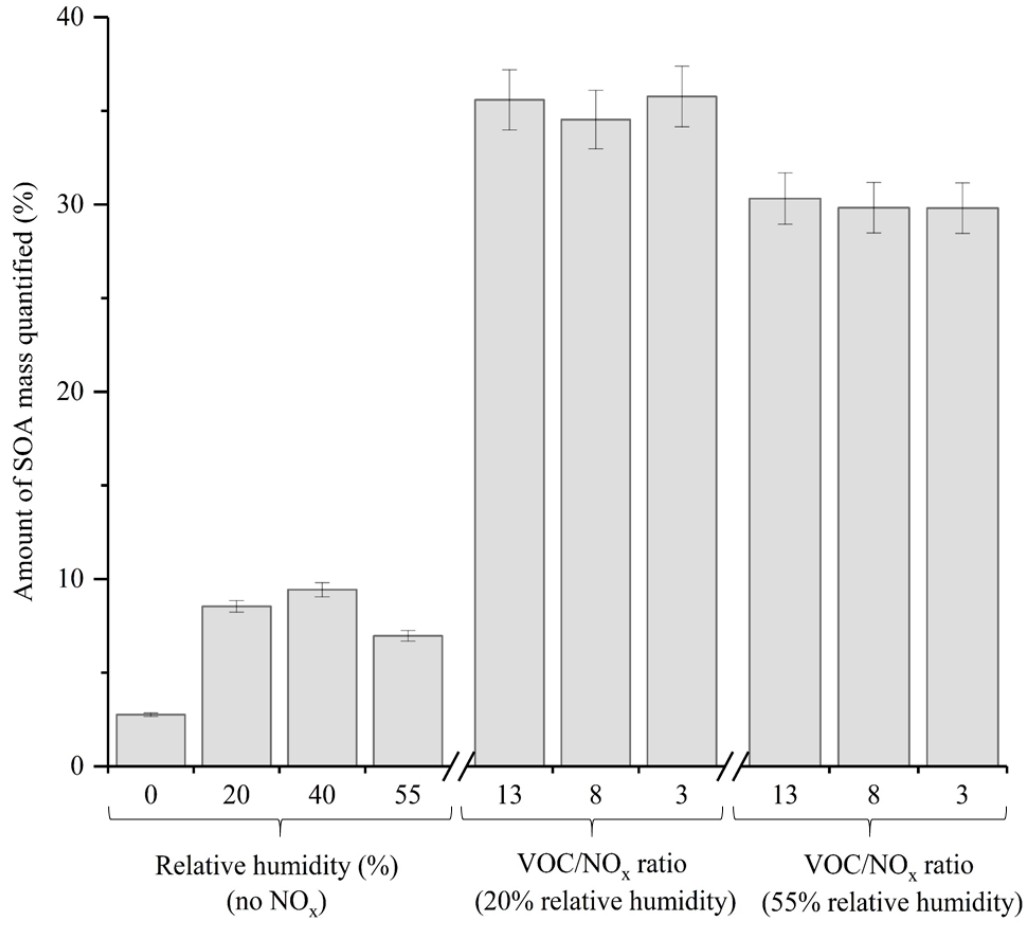

**Figure 7** – Total amount of SOA mass quantified using the generated standards in the α-pinene experiments shown in Table
1. Error bars represent the propagated uncertainty in the slope of the calibrations used to quantify each SOA component.





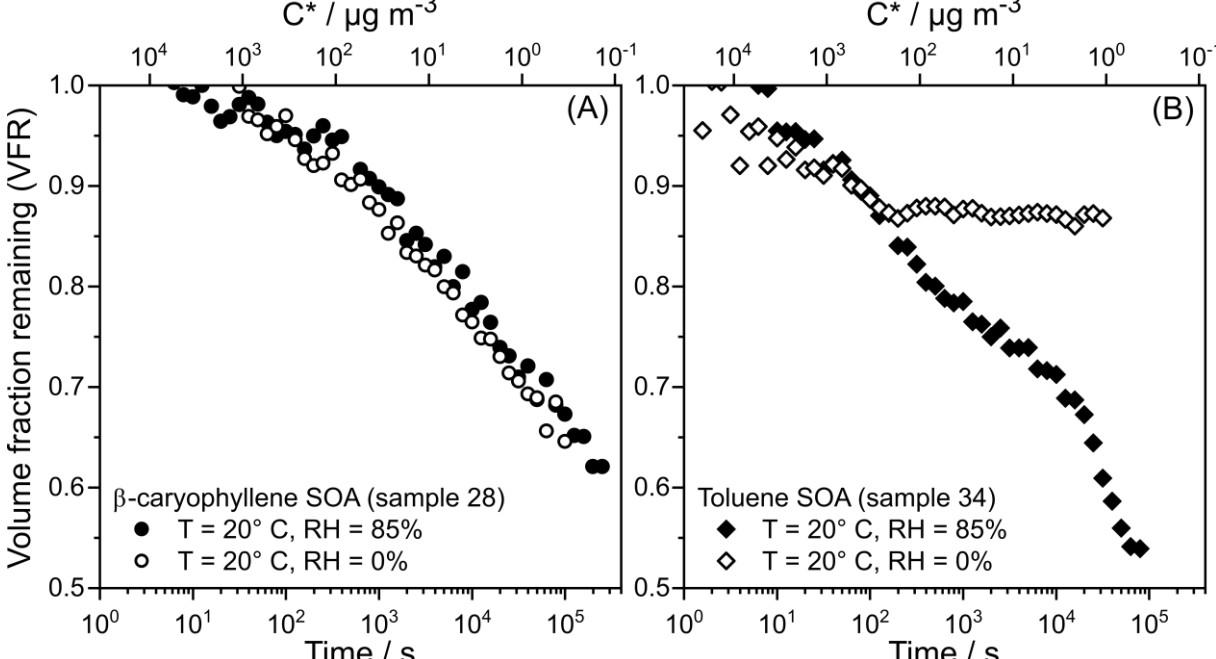

**Figure 8** – Volume fraction remaining (VFR) for single β-caryophyllene SOA (A) and toluene SOA (B) droplets confined in an electrodynamic balance (EDB) over ~ 1 day. VFR is compared for evaporation into high RH (solid symbols) and dry conditions (open symbols). Sample numbers correspond to the experiments shown in Table 1. Secondary *x*-axis displays the calculated effective saturation concentration ($C^*$, µg m$^{-3}$), see text for further information.