# Peer review of "A New Aerosol Flow Reactor to Study Secondary Organic Aerosol"

_Atmospheric Measurement Techniques, 2019_

## Referee Comment (RC1) · Anonymous Referee #3 · 8 Mar 2019

General Comments

This paper describes the design and operation of a new continuous flow reactor (CFR) for investigating the chemical composition and physical properties of secondary organic aerosol (SOA). The reactor was used to generate SOA from the photo-oxidation of four different precursors under a variety of experimental conditions. The SOA was collected onto filters or impactor plates and the chemical composition and physical properties were investigated using a range of off-line analytical techniques. The main advantage of this experimental apparatus is that it allows production of significantly more SOA mass than typically generated in simulation chamber experiments, thus making off-line analytical techniques more accessible. Indeed, sufficient SOA was generated in the experiments to enable CHNS elemental analysis to be performed, a technique which

is rarely performed on SOA samples.

Overall, this is a very well written paper which describes a useful new facility for generating SOA for off-line analysis. Technical aspects of the design, testing and operation of the CFR are described with a high level of detail. The results from the test experiments are of high quality and are well presented and interpreted. A more detailed analysis of SOA chemical composition will be presented in a future publication. One important finding of this work is the observed discrepancy in O/C ratio when measured by CHNS analysis and the more commonly applied technique of UHRMS. Further work is required in this area.

The CFR and analytical approaches presented in this paper represent a welcome addition to the range of experimental methods for investigating the complex nature of SOA. Publication in Atmospheric Measurement Techniques is recommended following consideration of the minor comments below.

Minor Comments

1. What is the difference between oxidative flow reactors (briefly described in the Introduction) and the continuous flow reactor built and operated by the authors? What is unconventional about the way the CFR is being used here? These points need to be clarified somewhere in the manuscript, probably in the last paragraph of the Introduction.

2. The light source emits radiation at 254 nm and 185 nm and is not representative of tropospheric conditions. While the authors do comment that the CFR is not being used to mimic atmospheric conditions, it is still important to ensure that the higher energy UV light used in these experiments does not significantly affect the representativeness of the oxidation chemistry of the SOA precursor, or the composition of the SOA itself. Maybe this issue can be addressed in section 3.3 CFR limitations?

3. Since the experiments were performed using high concentrations of precursors and

nitrogen oxides, there is the strong possibility of artefacts caused by deposition of gas-phase organic species on the filters and impactor plates. There is also the possibility of reactive nitrogen species interacting with the SOA via heterogeneous processes. The authors should provide some comments on the issue of possible artefacts. Denuders are commonly used in chamber experiments to remove gas-phase oxidation products and reduce artefacts. Could they be used in this set-up?

4. Page 8, lines 16-17: The SOA mass and number concentrations in the chamber background experiments given here seem to be very high (compared to chamber experiments). How do these concentrations compare., e.g. in % terms, to the concentrations produced during an experiment?

Technical comments

1. Page 1, line 16: Delete "mass"

2. Page 9, line 2: Replace "volatiles" with "organic compounds"

3. Page 14, line 3 and several other places in the manuscript and SI: The authors use the term "alcoholic hydroxyl" or "alcohol", whereas I think that simply "hydroxyl" is more appropriate.

4. Page 17, line 33: should be "affect"

5. Page 21, line 5: should be "affecting"

---

## Referee Comment (RC2) · Anonymous Referee #4 · 9 Mar 2019

This paper described the setup of a custom-built aerosol flow reactor, which was designed to generate large amount of secondary organic aerosols (above 10ˆ2 mg ) from different VOCs precursors using continuous flow mode. The RH, VOCs mixing ratio and VOCs/NOx condition can be controlled independently in the flow reactor. A series of offline analytical techniques was used to determine the chemical information of generated SOA including: CHNS elemental analyzer, nuclear magnetic resonance spectroscopy (NMR), ultra-performance liquid chromatography ultra-high resolution mass spectrometry (UHRMS) etc. Brief measurement results from those techniques were shown. After reading the paper, I have two major comments about the novelty and logic of this paper. Based on those comments, I recommend a major revision of this paper Major comments ïïjĹ1ïïjĽI was puzzled by the novelty of this continuous flow re-

actor (CFR), as titled in this paper. The common major advantages of CFR are (1) to achieve the intermediate NO chemical region as illustrated in Zhang et al. (Zhang et al., 2018) and (2) less wall losses, although similar wall losses between continuous flow-mode chamber and batch-mode chamber were found in some studies. The authors address the novelty of CFR in this study is its ability to generate much higher SOA concentrations compared to a normal batch-mode chamber. However, high SOA mass concentrations also can be achieved by many other types of already widely-used flow reactors e.g., commercialized oxidation flow reactor (or named potential aerosol mass flow reactor) (Kang et al., 2007) or some custom-built flow reactors e.g., (Huang et al., 2017). Additionally, the offline techniques mentioned in this study have already been applied to analyze SOA generated in normal chamber. E.g. two-dimensional heteronuclear NMR spectroscopy to (Maksymiuk et al., 2009), CHNS elemental analyzer (Kroll et al., 2011), HPLC-ITMS (Hamilton et al., 2011;Pereira et al., 2014) and Volatility measurement (Huffman et al., 2009). Those results suggest SOA formed from normal chamber with higher precursor concentrations and longer reacting times can also meet the detection limit of those offline techniques mentioned in this paper. If the advantage of CFR in this study is only to provide more SOA masses, I do not think it is a novel method.

ïijĹ2ïijĽ The title of the paper is "Novel Aerosol Flow Reactor to Study Secondary Organic Aerosol", whereas the authors did not really show much basic characterization information from this aerosol flow reactor. e.g., what the OH concentration (or OH exposure) ranges can be achieved in CFR, how much photon flux of lamps at different light settings (which is crucial of SOA photolysis), What are the wall losses. The measurement results from different techniques are not the characteristics of flow reactors. One or two measurement examples from those offline techniques should be enough if the story in this study is really to show the flow reactor.

Other comments: Page 7 Line 12-14, I did not see the difference of this CFR with the current used oxidation flow reactor e.g., (Kang et al., 2007;Lambe et al., 2011;Huang

et al., 2017). While emphasizing the merits of the CFR, could the authors show the advantage of this CFR compared to other flow reactors used in the lab and field studies. Page 7 line 30: dilution can be made to measure the NOx and VOCs. Page 7 Line 31-32: What kind of separate experiments were done ? Page 8 Line 16-18: SOA mass concentration was quite high for background concentrations in a chamber. Will these background SOA contaminate the newly formed SOA in new experiments? Page 8 line 32: What the OH concentration can be achieved in the CFR? Page 13 line 6: Was the reactor temperature controlled manually or only influenced by room temperature? The description in this sentence was not consistent with the actually measured temperatures listed in the Table 1. Page 13 line 17: Unit should be added Page 14 line 31-32: Without considering OH exposure in the CFR during different experiments, the oxidation state of different type of SOAs vary significantly. I do not think the O/C range reported from different experiments can be used to support the accuracy of CHNS method.

Page 15 line 8-11: Similarly, OH exposure should be considered. And much higher H/C ratios of a-pinene SOA from CFR were found in Fig. 5 compared to literature results.

---

## Referee Comment (RC3) · Anonymous Referee #1 · 13 Mar 2019

Pereira et al. presented a chamber operated in continuous mode (CFR) with conceptually the same photochemistry initiating method as the "OFR185" operation mode of oxidation flow reactors (OFRs) (George et al., 2007; Kang et al., 2007; Lambe et al., 2011; Li et al., 2015). They used their CFR to produce very large amounts of SOA for a suite of offline physicochemical analyses, including some requiring high OA amounts. Although the inlet was well controlled and the offline analysis was comprehensive, the reactor and experiment design have a couple of fundamental problems. These are so major that it is unclear to me whether the paper should be published, unless the issues are described thoroughly and the paper used to describe an incremental design step that was not quite successful, which will be built upon to achieve a more atmospherically-relevant reactor in a future iteration.

(1) Although the volume being larger than OFRs is touted as an advantage of the CFR and part of its novelty, this apparent advantage is negated by the distribution of UV light and species of interest, which appears to be extremely heterogeneous. A pen-ray was used as the light source in the CFR. Although the authors did not specify the dimension of its lighted area in the paper, I searched for this information on the website of its manufacturer ([https://www.uvp.com/mercury](https://www.uvp.com/mercury)) and it would appear that it is very small (lighted length as small as <2 cm). Even if it is slightly larger than that, the pen-ray can be roughly regarded as a point light source given the large volume of the CFR. Then the photon flux scales inversely with the square of the distance to the light source. Let's assume that a spherical UV source with a diameter of 5 cm (much larger surface area than a stick-like lamp with a lighted length of 5 cm). Then there is only 1% of the initial UV intensity (next to the lamp) only ~20 cm from the pen-ray surface under the assumption of no light absorption, and only 0.08% of the initial intensity near the corners of the reactor (assuming the UV light placed right in the middle of the reactor). The UV absorption at 185 nm by $O_2$ exacerbates this problem. $O_2$, with a cross section of ~$10^{-20}$ $cm^2$ at 185 nm, only needs a ~20 cm optical path to reach an optical depth of 1. This leads to an additional e-fold decay of the intensity every 20 cm, *in addition* to the intensity decay caused by the geometry. When applying this effect, the light remaining at 20 cm and the reactor corners is 0.6% and 0.0008% of the initial values, respectively. 99.5% (93%) of the reactor volume has light intensities smaller than those near the light by a factor of 10 (100).

[Figure]

Figure: Contour plot of the estimated field of 185 nm UV intensity relative to that at the lamp surface in the horizontal section cutting the center of the lamp, under the assumptions of i) a spherical UV lamp with a diameter of 5 cm ii) placed in the center of the CFR, iii) $O_2$ cross section at 185 nm of $1x10^{-20}$ $cm^2$ and 1 atm of air pressure, and iv) absorption of the 185 nm light by the reactor surfaces. The lamp center is set as (0, 0).

Therefore, despite its larger volume than most OFRs, most of the volume in the CFR seems to be photochemically "dark" and its photochemically useful volume is actually smaller than common OFRs, e.g., PAM and CPOT (in almost the whole internal space of PAM (volume: ~13 L) for all commonly used lamp placements, the relative 185 UV intensity to the lamp surface is >5% (Peng et al., 2018), while for the case discussed above, the volume with a relative 185 nm UV>5% is only ~3.5 L). Thus despite its large volume, this is a very small effective reactor.

Even within the photochemically active space, UV intensity still varies substantially, which makes it difficult to relate offline analysis results to a certain reaction condition. Then the results of the offline analysis are less informative.

I suggest that in the future the authors use multiple lights in a better layout to make the UV field as uniform as possible. This would lead to more uniform conditions and the production of more meaningful SOA material.

(2) The experiments were conducted using tens of ppm of VOC and several ppm of NOx, corresponding to OH reactivities of thousands of $s^{-1}$ and more (ranging from ~3,000 $s^{-1}$ for Exps. 34 and 35 to ~300,000 $s^{-1}$ for Exps. 26-28). These extraordinarily high reactivities are certain to reduce OH concentration in the CFR by several orders of magnitude (Peng et al., 2015). But UV intensity at 185 and 254 nm is not reduced by the addition of the VOCs, and could play a major role in VOC loss compared to reactions with OH (Peng et al., 2016), especially for toluene, which strongly absorbs at 185 and 254 nm.

[Figure]

Figure: plot to quantify the relative importance of 185 nm VOC photolysis to their reactions with OH in same format as Fig. 1 of Peng et al. (2016). The x-axis positions of Exps. 9 and 38 are estimated using the OFR Exposure Estimator (https://sites.google.com/site/pamwiki/hardware/estimation-equations) under the assumption of a uniform 185 nm UV of $1\times10^{13}$ ph $cm^{-2}$ $s^{-1}$ in the CFR. That of Exp. 28 has also been estimated. Its value is too large (~$1\times10^{7}$ cm/s) to be shown in the plot, and would be

expected to be so large that many molecules would have their fate impacted or dominated by 185 nm photolysis.

Energetic 185 and 254 nm photons may result in a very different organic radical chemistry than in the atmosphere and typical chamber experiments. Although the authors claimed that their objective was not to perfectly mimic atmospheric conditions, clearly a key goal is to produce SOA that is atmospherically-relevant. For example, they repeatedly compared their offline analysis results to ambient measurement and chamber experiment results in the literature as validations of their experiments. If the CFR was only to produce SOA to test several offline analytical instruments with a complex mixture of oxidized species, the current CFR experimental design is purely a laboratory exercise and viable as such. If the authors assume that their CFR-produced SOA may serve as surrogate of ambient and/or typical chamber SOA to any extent (even though the experimental conditions do not replicate ambient conditions), the unrealistic photochemistry initiated by 185 and 254 nm UV should be avoided. In the experiments reported in this paper, OH was always substantially reduced by VOC and $NO_x$. Thus most of the SOA samples shown in Fig. 5 were only weakly oxidized. Toluene-derived SOA was an exception because strong photolysis at 185 and 254 nm may have produced more organic radicals (followed by $O_2$ addition etc., leading to higher O:C). Those photolysis products were likely to be smaller and more volatile, and have lower SOA yields. The peculiarity of toluene experiments suggests the importance of strong 185 and 254 nm VOC photolysis occurring.

If the authors intend to claim any relevance of CFR-produced SOA to ambient and/or typical chamber SOA, they have to limit the amount of VOC (and $NO_x$) injected to avoid the above problem. OH reactivity of tens of $s^{-1}$ has been recommended for similar reactors using the same OFR185 photochemistry, in order to maintain the chemistry in a tropospherically-relevant regime (Peng et al., 2016; Peng and Jimenez, 2017). **This is 100-10000 times lower than the reactivities used in this paper.** Assuming a VOC reacting with OH at $10^{-11}$ $cm^3$ $molec^{-1}$ $s^{-1}$, roughly 100 ppb of VOC can be injected into the reactor without entering conditions with significant 185 or 254 nm VOC (and product) photolysis. Then this is roughly the upper limit of OA that can be made in OFRs (including the CFR) using Hg lamps to generate OH. **This is about 100 times lower than the concentrations used in this paper.**

In this sense, if the authors do not employ multiple lights to largely make use of the volume of the CFR, its SOA production capacity is not superior to other OFRs (e.g., PAM). There have already been a number of papers where other OFRs were used to produce SOA that was deposited on substrates and collected on filters, for SFG and viscosity analysis, respectively (e.g., Shrestha et al., 2015; Song et al., 2016). Besides, PEAR, a large OFR, has been recently presented and appears to have a more appropriate design for the purpose of producing SOA in large quantities (Ihalainen et al., 2018). Thus I do not think that OFR (CFR) as a tool to produce larger concentrations of SOA (by collecting over a longer period of time) is really a novel concept.

**Specific comments:**

Page 3, Line 30: OFRs (e.g., Aerodyne PAM) also have good flow and precursor injection control.

Table 1: there was no really low-$NO_x$ experiments among those with $NO_x$ injected in Table 1. Even with a VOC:$NO_x$ ratio of 13, NOx was still injected in ppm. Compared to $HO_2$ (not VOC), $NO_x$ should have always dominated $RO_2$ fate in the CFR experiments reported in this paper.

Section 2.3: although offline analysis methods are not the main focus of this paper, a brief description of potential artifacts in these offline analyses would still be helpful.

**Technical corrections:**

Figure 1: please change the bag volume from "3 $m^3$" to "0.3 $m^3$".

Page 14, Line 5 and Page 19, Line 10: references Cao et al. and Shrivastava et al. are missing in the reference list.

Page 14, Line 7: please add "of" after "intensity".

Page 19, Line 33: is "Although a faster evaporation rate…" a part of the preceding sentence?

Page 20, Line 10: the word "bin" is missing after "µg m$^{-3}$".

**References:**

George, I. J., Vlasenko, A., Slowik, J. G., Broekhuizen, K. and Abbatt, J. P. D.: Heterogeneous oxidation of saturated organic aerosols by hydroxyl radicals: uptake kinetics, condensed-phase products, and particle size change, Atmos. Chem. Phys., 7(16), 4187–4201, doi:10.5194/acp-7-4187-2007, 2007.

Ihalainen, M., Tiitta, P., Czech, H., Yli-Pirilä, P., Hartikainen, A., Kortelainen, M., Tissari, J., Stengel, B., Sklorz, M., Suhonen, H., Lamberg, H., Leskinen, A., Kiendler-Scharr, A., Harndorf, H., Zimmermann, R., Jokiniemi, J. and Sippula, O.: A novel high-volume Photochemical Emission Aging flow tube Reactor (PEAR), Aerosol Sci. Technol., 0(0), 1–19, doi:10.1080/02786826.2018.1559918, 2018.

Kang, E., Root, M. J., Toohey, D. W. and Brune, W. H.: Introducing the concept of Potential Aerosol Mass (PAM), Atmos. Chem. Phys., 7(22), 5727–5744, doi:10.5194/acp-7-5727-2007, 2007.

Lambe, A. T., Ahern, A. T., Williams, L. R., Slowik, J. G., Wong, J. P. S., Abbatt, J. P. D., Brune, W. H., Ng, N. L., Wright, J. P., Croasdale, D. R., Worsnop, D. R., Davidovits, P. and Onasch, T. B.: Characterization of aerosol photooxidation flow reactors: heterogeneous oxidation, secondary organic aerosol formation and cloud condensation nuclei activity measurements, Atmos. Meas. Tech., 4(3), 445–461, doi:10.5194/amt-4-445-2011, 2011.

Li, R., Palm, B. B., Ortega, A. M., Hu, W., Peng, Z., Day, D. A., Knote, C., Brune, W. H., de Gouw, J. and Jimenez, J. L.: Modeling the radical chemistry in an Oxidation Flow Reactor (OFR): radical formation and recycling, sensitivities, and OH exposure estimation equation, J. Phys. Chem. A, 119(19), 4418–4432, doi:10.1021/jp509534k, 2015.

Peng, Z., Day, D. A., Ortega, A. M., Palm, B. B., Hu, W., Stark, H., Li, R., Tsigaridis, K., Brune, W. H. and Jimenez, J. L.: Non-OH chemistry in oxidation flow reactors for the study of atmospheric chemistry systematically examined by modeling, Atmos. Chem. Phys., 16(7), 4283–4305, doi:10.5194/acp-16-4283-2016, 2016.

Peng, Z., Day, D. A., Stark, H., Li, R., Lee-Taylor, J., Palm, B. B., Brune, W. H. and Jimenez, J. L.: HOx

radical chemistry in oxidation flow reactors with low-pressure mercury lamps systematically examined by modeling, Atmos. Meas. Tech., 8(11), 4863–4890, doi:10.5194/amt-8-4863-2015, 2015.

Peng, Z. and Jimenez, J. L.: Modeling of the chemistry in oxidation flow reactors with high initial NO, Atmos. Chem. Phys., 17(19), 11991–12010, doi:10.5194/acp-17-11991-2017, 2017.

Peng, Z., Palm, B. B., Day, D. A., Talukdar, R. K., Hu, W., Lambe, A. T., Brune, W. H. and Jimenez, J. L.: Model Evaluation of New Techniques for Maintaining High-NO Conditions in Oxidation Flow Reactors for the Study of OH-Initiated Atmospheric Chemistry, ACS Earth Sp. Chem., 2(2), 72–86, doi:10.1021/acsearthspacechem.7b00070, 2018.

Shrestha, M., Zhang, Y., Upshur, M. A., Liu, P., Blair, S. L., Wang, H., Nizkorodov, S. A., Thomson, R. J., Martin, S. T. and Geiger, F. M.: On Surface Order and Disorder of α-Pinene-Derived Secondary Organic Material, J. Phys. Chem. A, 119(19), 4609–4617, doi:10.1021/jp510780e, 2015.

Song, M., Liu, P. F., Hanna, S. J., Zaveri, R. A., Potter, K., You, Y., Martin, S. T. and Bertram, A. K.: Relative humidity-dependent viscosity of secondary organic material from toluene photo-oxidation and possible implications for organic particulate matter over megacities, Atmos. Chem. Phys., 16(14), 8817–8830, doi:10.5194/acp-16-8817-2016, 2016.

---

## Author Comment (AC1) · 22 May 2019

The authors would like to thank the reviewers for their comments. Please find our response below (highlighted in blue). Please note, due to the similarity in the reviewers' comments, we have uploaded our response to all reviewers in one document rather than separating our responses.

**Referee 3**

This paper describes the design and operation of a new continuous flow reactor (CFR) for investigating the chemical composition and physical properties of secondary organic aerosol (SOA). The reactor was used to generate SOA from the photo-oxidation of four different precursors under a variety of experimental conditions. The SOA was collected onto filters or impactor plates and the chemical composition and physical properties were investigated using a range of off-line analytical techniques. The main advantage of this experimental apparatus is that it allows production of significantly more SOA mass than typically generated in simulation chamber experiments, thus making off-line analytical techniques more accessible. Indeed, sufficient SOA was generated in the experiments to enable CHNS elemental analysis to be performed, a technique which is rarely performed on SOA samples.

Overall, this is a very well written paper which describes a useful new facility for generating SOA for off-line analysis. Technical aspects of the design, testing and operation of the CFR are described with a high level of detail. The results from the test experiments are of high quality and are well presented and interpreted. A more detailed analysis of SOA chemical composition will be presented in a future publication. One important finding of this work is the observed discrepancy in O/C ratio when measured by CHNS analysis and the more commonly applied technique of UHRMS. Further work is required in this area. The CFR and analytical approaches presented in this paper represent a welcome addition to the range of experimental methods for investigating the complex nature of SOA. Publication in Atmospheric Measurement Techniques is recommended following consideration of the minor comments below.

**Minor Comments:**

1. What is the difference between oxidative flow reactors (briefly described in the Introduction) and the continuous flow reactor built and operated by the authors? What is unconventional about the way the CFR is being used here? These points need to be clarified somewhere in the manuscript, probably in the last paragraph of the Introduction.

The CFR has a considerably larger volume (0.3 m$^3$ *vs.* ~ 0.001 to 0.01m$^3$) and a longer residence time (greater than ~ 25 mins *vs.* seconds to a few minutes) than typical oxidative flow reactors, increasing the amount of SOA which can be formed in the reactor. The longer residence times used in the CFR will also allow the generated SOA more time to achieve equilibrium with the gas-phase, in comparison to reactors with residences times less than a few hundred seconds. The CFR has also been designed to allow the rapid replacement of the reactor sampling bag at minimal cost, considerably reducing the reactor cleaning time in comparison to other reactors which are constructed from stainless-steel or glass. We have added the following into the manuscript (see page 3, line 31), "VOCs and oxidants are continuously introduced into the reactor and sample air extracted, operating under steady-state flow conditions (analogous to oxidative flow reactors), allowing a wide range of chemical scenarios to be investigated through the control of reactant mixing ratios and flow rates (*i.e.* residence time). In contrast to oxidative flow reactors, the developed CFR has a considerably larger volume (0.3 m$^3$ *vs.* ~ 0.001 to 0.01m$^3$) and longer residence times (greater than ~ 25 mins *vs.* seconds to a few minutes), increasing the amount of SOA which can be formed in the reactor. Furthermore, the longer residence times used in the CFR will potentially allow the generated SOA more time to achieve equilibrium with the gas-phase, in comparison to reactors with residences times less than a few hundred seconds (*e.g.* Anttila et al. (2016)). High VOC and oxidant mixing ratios (*i.e.* ppmv levels) are used in this study to generate large quantities of SOA mass. Consequently, the CFR has been designed to allow the reactor sampling bag to be rapidly replaced and at minimal cost, significantly reducing reactor cleaning time in comparison to oxidative flow reactors which are constructed out of stainless-steel or glass (*e.g.* (Huang et al., 2017; Ihalainen et al., 2019))."

Our unconventional use is attributed to the high VOC and oxidant mixing ratios (*i.e.* ppmv levels) used in this study to generate large quantities of SOA mass. This is not a common approach, with many studies focusing on generating SOA using near ambient mixing ratios. We have included the following (see page 3, line 29), "In contrast to majority of atmospheric simulation chamber and reactor studies, we show how generating large quantities of SOA mass (> 10$^2$ mg per experiment) which is usually avoided, can be used to gain greater insights into the complex physiochemical properties controlling gas-particle partitioning."

2. The light source emits radiation at 254 nm and 185 nm and is not representative of tropospheric conditions. While the authors do comment that the CFR is not being used to mimic atmospheric conditions, it is still important to ensure that the higher energy UV light used in these experiments does not significantly affect the representativeness of the oxidation chemistry of the SOA precursor, or the composition of the SOA itself. Maybe this issue can be addressed in section 3.3 CFR limitations?

We have addressed this issue in section 3.3 CFR limitations (page 22, line 11) which also includes our response to Referee 5.

We have added the following into the manuscript, "The UV lamps used in the CFR had light emissions with wavelengths at 254 nm (primary energy) and 185 nm. The 185 nm wavelength may result in very different organic radical chemistry than observed in the ambient atmosphere (*e.g.* see Peng et al. (2016) for further information), potentially affecting the observed SOA composition. In addition, the light intensity emitted from the UV lamps was not sufficient to provide uniform light distribution within the reactor. It is strongly recommended that the UV light source is modified in future studies, including multiple UV lamps (increasing the light distribution within the reactor) which do not emit a 185 nm wavelength. It must be stressed however, that the objective of this study was to investigate the effect of chemical composition on the physical state of the generated SOA, furthering our understanding of the physicochemical relationship/s controlling gas-particle partitioning. These physiochemical relationship/s are determined by the chemical and physical properties of each SOA sample and are not affected by the atmospheric relevance of generated SOA."

3. Since the experiments were performed using high concentrations of precursors and nitrogen oxides, there is the strong possibility of artefacts caused by deposition of gas-phase organic species on the filters and impactor plates. There is also the possibility of reactive nitrogen species interacting with the SOA via heterogeneous processes. The authors should provide some comments on the issue of possible artefacts. Denuders are commonly used in chamber experiments to remove gas-phase oxidation products and reduce artefacts. Could they be used in this set-up?

Gas-phase adsorption to the impactor plates of the ELPI is negligible due to its design, *i.e.* particles are impacted onto size segregated impactor plates (based on their aerodynamic size) whilst under a strong low vacuum which continuously removes gas-phase species. Denuders or an activated charcoal trap could be used prior to the quartz fibre filter to prevent gas-phase absorption. We have added the following into the manuscript (page 21, line 27), "The CFR was designed as a simple, low-cost tool to generate large quantities of SOA mass for offline composition and single particle analysis. The high VOC and oxidant mixing ratios (*i.e.* ppmv levels) used in study may increase the possibility of reactive nitrogen species interacting with the SOA *via* heterogenous processes, affecting the observed SOA chemical composition (*e.g.*(Montoya-Aguilera et al., 2018)). Furthermore, there is a strong possibility of artefacts from gas-phase adsorption to the quartz fibre filters (Parshintsev et al., 2011). Gas-phase adsorption to the ELPI is negligible due to its design (*i.e.* particles are collected onto size segregated impactor plates (based on their aerodynamic size) whilst under a strong low vacuum which continuously removes gas-phase species). All compositional and single particle analysis techniques were performed on the SOA collected from the ELPI, with the exception of infra-red spectroscopy, which was performed on the SOA collected onto the quartz fibre filters. Thus, it is possible that the quartz fibre filters analysed using infra-red spectroscopy may be affected by artefacts. Future studies should use an activated charcoal trap prior to the quartz fibre filter to prevent gas-phase absorption. The offline techniques used in this study are unlikely to introduce a major source of artefacts into the samples, providing instrument background or blanks runs are performed and the contaminants subtracted from the sample data, as performed in this work. Artefacts are more commonly introduced into the samples through preparation methods (*e.g.* filter extraction processes) for analysis using offline techniques. The use of the ELPI minimised the potential introduction of artefacts into the samples through the exclusion of all extraction processes, *i.e.* samples were either analysed without modification or dissolved into high purity solvents (without temperature or pressure changes)."

4. Page 8, lines 16-17: The SOA mass and number concentrations in the chamber background experiments given here seem to be very high (compared to chamber experiments). How do these concentrations compare., e.g. in % terms, to the concentrations produced during an experiment?

The SOA mass formed in the chamber background experiments represented < 3.2 % of the SOA mass formed in the α-pinene low mixing ratio experiments (exp. 7 and 14) and < 1.1 % in all other experiments. Particle number concentrations in the chamber background experiments represented $11.0 \pm 12.5\%$ (arithmetic mean ± relative standard deviation, shown in percentage) of the average number of particles formed in the α-pinene,

limonene, β-caryophyllene and toluene experiments, excluding experiments 2 and 17. Experiments 2 and 17 were observed to have lower particulate number concentrations than observed in the chamber background experiments, the reason for which is unclear. We have added the above into the manuscript, please see page 8, line 32.

**Technical comments:**

1. Page 1, line 16: Delete "mass"

Removed.

2. Page 9, line 2: Replace "volatiles" with "organic compounds"

Changed.

3. Page 14, line 3 and several other places in the manuscript and SI: The authors use the term "alcoholic hydroxyl" or "alcohol", whereas I think that simply "hydroxyl" is more appropriate.

Changed.

4. Page 17, line 33: should be "affect"

Changed.

5. Page 21, line 5: should be "affecting"

Changed.

**Referee 4**

This paper described the setup of a custom-built aerosol flow reactor, which was designed to generate large amount of secondary organic aerosols (above $10^2$ mg) from different VOCs precursors using continuous flow mode. The RH, VOCs mixing ratio and VOCs/NO$_x$ condition can be controlled independently in the flow reactor. A series of offline analytical techniques was used to determine the chemical information of generated SOA including: CHNS elemental analyzer, nuclear magnetic resonance spectroscopy (NMR), ultra-performance liquid chromatography ultra-high resolution mass spectrometry (UHRMS) etc. Brief measurement results from those techniques were shown. After reading the paper, I have two major comments about the novelty and logic of this paper. Based on those comments, I recommend a major revision of this paper.

**Major comments**

1. I was puzzled by the novelty of this continuous flow reactor (CFR), as titled in this paper. The common major advantages of CFR are (1) to achieve the intermediate NO chemical region as illustrated in Zhang et al. (Zhang et al., 2018) and (2) less wall losses, although similar wall losses between continuous flow-mode chamber and batch-mode chamber were found in some studies. The authors address the novelty of CFR in this study is its ability to generate much higher SOA concentrations compared to a normal batch-mode chamber. However, high SOA mass concentrations also can be achieved by many other types of already widely-used flow reactors e.g., commercialized oxidation flow reactor (or named potential aerosol mass flow reactor) (Kang et al., 2007) or some custom-built flow reactors e.g., (Huang et al., 2017). Additionally, the offline techniques mentioned in this study have already been applied to analyze SOA generated in normal chamber. E.g. two-dimensional heteronuclear NMR spectroscopy to (Maksymiuk et al., 2009), CHNS elemental analyzer (Krolletal., 2011), HPLC-ITMS (Hamilton et al., 2011; Pereira et al.,2014) and volatility measurement (Huffman et al., 2009). Those results suggest SOA formed from normal chamber with higher precursor concentrations and longer reacting times can also meet the detection limit of those offline techniques mentioned in this paper. If the advantage of CFR in this study is only to provide more SOA masses, I do not think it is a novel method.

The use of "novel" referred to our approach, rather than being solely attributed to the CFR design. We have changed the title of our manuscript to make this clearer. The title now reads "A New Aerosol Flow Reactor to Study Secondary Organic Aerosol". There are several similarities between the major comments here and those by Referee 5. Please also see our response to Referee 5. The novelty of this study centres on the uncharacteristic use of the developed CFR to generate large quantities of SOA mass, allowing us to use highly accurate analytical techniques (which are usually inaccessible due to the large quantities of SOA mass required for analysis) to investigate the chemical and physical properties of each generated SOA sample. Many studies

focus on generating SOA using near ambient mixing ratios. Here, we show how generating considerable quantities of SOA mass (> $10^2$ mg per experiment) which is usually avoided, can be used to evaluate the accuracy of commonly used techniques (*i.e.* UHRMS), generate non-commercially available standards for SOA quantification and gain further insights into the complex physiochemical properties controlling SOA dynamics. The key point here, is that all of the techniques listed in this study have been used to investigate the chemical and physical properties of each generated SOA sample. There are no studies which have performed such a comprehensive set of measurements, possible only because of the large amount of SOA mass which can be generated using the CFR. The studies cited above (*e.g.* Maksymiuk et al., 2009; Krolletal., 2011; Hamilton et al., 2011; Pereira et al.,2014; Huffman et al., 2009) use one or two analytical techniques, some which are very common analysis methods (*i.e.* HPLC-ITMS). Techniques such as the electrodynamic balance, CHNS elemental analyser (see Referee 3's summary), two-dimensional NMR spectroscopy and semi-preparative liquid chromatography mass spectrometry for the generation of non-commercially available standards, are rarely used within aerosol science because of the amount of SOA mass required for analysis, which cannot be generated in "normal" batch mode chamber experiments. For example, Maksymiuk et al. 2009 is one of approximately 5 studies which have used two-dimensional NMR to investigate SOA composition (investigated limonene SOA). To our knowledge, no studies have used two-dimensional NMR spectroscopy to investigate the chemical composition of toluene and β-caryophyllene SOA, as shown in this work. We also present the first use of an electrodynamic balance to assess the influence of the temperature and phase state of the SOA on the volatilisation kinetics of semi-volatile components from a sample particle. Thus, the novelty of this work does not centre solely on the design of the CFR, but the additional highly accurate compositional and physical state measurements which can be obtained using this methodology.

It is unclear whether the PAM reactor is capable of generating > $10^2$ mg of SOA mass per experiment. The PAM reactor has a considerably smaller volume (0.013 $m^3$ *vs.* 0.3 $m^3$) and shorter residence time (~ 80 s, (Zhang et. al (2018) *vs.* greater than ~ 25 mins) and would most likely need to be operated over longer time periods (*e.g.* several days) to generate the same amount of SOA mass per experiment. Furthermore, the SOA generated in the PAM reactor is unlikely to be in equilibrium with the gas phase due to the short residence times (Anttila et al. (2016)). The reactor developed by Huang et al. (2017) also has a smaller volume of 0.03 $m^3$ (factor of 10 smaller than the CFR) and is constructed out of glass. One of the main advantages of the CFR, is the ability to rapidly change the reactor sampling bag. Reactors which are constructed out of material which are not designed to be easily replaced (*e.g.* glass, stainless-steel) will require considerable cleaning and are more likely to exhibit "memory effects" from the high mixing ratios (*i.e.* ppmv levels) required for this work. In addition, we assume the PAM reactor and the reactor developed in Huang et al (2017) cost more to build than the CFR developed in this study (cost = ~ £8000).

2. The title of the paper is "Novel Aerosol Flow Reactor to Study Secondary Organic Aerosol", whereas the authors did not really show much basic characterization information from this aerosol flow reactor. e.g., what the OH concentration (or OH exposure) ranges can be achieved in CFR, how much photon flux of lamps at different light settings (which is crucial of SOA photolysis), What are the wall losses. The measurement results from different techniques are not the characteristics of flow reactors. One or two measurement examples from those offline techniques should be enough if the story in this study is really to show the flow reactor.

The novelty of this work corresponds to the methodological approach rather than being solely attributed to the CFR design (see above comments). Thus, it is imperative to show both the design of the reactor and the types of results which can be obtained using the comprehensive suite of offline techniques presented in this work. The CFR was designed as a simple, low-cost tool to generate large quantities of SOA mass for offline compositional and physical state measurements. As a result, the CFR was not characterised (in this study) as extensively as other well established reactors. We must stress however, that the aim of this study was to investigate the effect of chemical composition on the physical state of the SOA, allowing us to further investigate the physicochemical relationships controlling gas-particle partitioning. These physicochemical relationships are determined by the chemical and physical properties of the generated SOA and are not affected by atmospheric relevance. However, we do agree that further characterisation of the CFR should be performed in future studies.

**Other comments:**

Page 7 Line 12-14, I did not see the difference of this CFR with the current used oxidation flow reactor e.g., (Kang et al., 2007; Lambe et al., 2011; Huang et al., 2017). While emphasizing the merits of the CFR, could the authors show the advantage of this CFR compared to other flow reactors used in the lab and field studies.

This has been added into the manuscript, please see page 3, line 31 and our response to Referee 3, minor comment 1.

Page 7 line 30: dilution can be made to measure the $NO_x$ and VOCs.

The sentence has been reworded, replacing "…could not be measured using..." for "…were not measured using…". Please see page 8, line 8.

Page 7 Line 31-32: What kind of separate experiments were done?

The following has been added into the manuscript (page 8 line 11), "α-pinene and NO mixing ratios were individually measured in two separate experiments. Both experiments were performed in the dark at 55 % relative humidity, using the same total reactor flow rate (*i.e.* residence time) as the experiments shown in Table 1. No other oxidants or VOCs were introduced into reactor during these experiments."

Page 8 Line 16-18: SOA mass concentration was quite high for background concentrations in a chamber. Will these background SOA contaminate the newly formed SOA in new experiments?

The SOA mass formed in the chamber background experiments represented < 3.2 % of the SOA mass formed in the two α-pinene low mixing ratio experiments (exp. 7 and 14) and < 1.1 % in all other experiments. The SOA mass formed in the chamber background experiments represented a small proportion of the SOA mass formed in the α-pinene, limonene, β-caryophyllene and toluene experiments and is thus likely to have a negligible effect on the generated SOA. Please see our response to Referee 3, minor comment 4 and page 8, line 32 in the manuscript.

Page 8 line 32: What the OH concentration can be achieved in the CFR?

The OH concentration was not measured in the reactor. Please see our response to major comment 2 (shown above).

Page 13 line 6: Was the reactor temperature controlled manually or only influenced by room temperature? The description in this sentence was not consistent with the actually measured temperatures listed in the Table 1.

The reactor temperature was influenced by room temperature and the heat generated from the UV lamp during operation. The room temperature was thermostatically controlled at 21°C. The reactor temperature increased upon UV irradiation and stabilised during each experiment, please see Page 13, line 28 and Figure 2. We have added the word "relatively" into the sentence, reading "…relatively stable…", to more accurately reflect our average experimental temperature and variation (*i.e.* 24.1 ± 1.0 °C), please see page 13 line 24.

Page 13 line 17: Unit should be added

Added.

Page 14 line 31-32: Without considering OH exposure in the CFR during different experiments, the oxidation state of different type of SOAs vary significantly. I do not think the O/C range reported from different experiments can be used to support the accuracy of CHNS method.

The accuracy of the CHNS elemental analysis has not been determined by the use of different VOCs or experimental conditions. Each SOA sample and crucially, the same sample, was analysed using both techniques (*i.e.* ultra-high resolution mass spectrometry and CHNS elemental analysis) allowing a direct comparison of elemental compositions obtained from each technique for each SOA sample. The CHNS elemental analyser is the more accurate technique in comparison to ultra-high resolution mass spectrometry. The CHNS elemental analyser determines the elemental composition of the sample using combustion (observes all carbon, hydrogen, nitrogen and sulfur present in the sample), whereas ultra-high resolution mass spectrometry uses a selective ionisation source (*i.e.* electrospray ionisation) where only a proportion of the sample is observed.

Page 15 line 8-11: Similarly, OH exposure should be considered. And much higher H/C ratios of a-pinene SOA from CFR were found in Fig. 5 compared to literature results.

We agree that OH exposure should be considered, please see our response to major comment 2. We do not agree that our α-pinene H/C ratios are much higher than the values reported in the literature. Zhao et al. (2015) reports numerous α-pinene H/C ratios from OH oxidation ranging between ~ 1.4 to 1.7 (please see https://www.atmos-chem-phys.net/15/991/2015/acp-15-991-2015.pdf, page 1004, Figure 6A). For comparison, our α-pinene SOA H/C ratios ranged from 1.57 to 1.67, with an average O/C ratio of 0.43.

Furthermore, the average α-pinene H/C ratio shown in Reinhardt et al. (2007) (estimated from their van Krevelen plot) was ~ 1.57. We report an α-pinene H/C ratio of 1.57 for one of our experiments, with the reminder of our experiments determined to have an H/C ratio within 0.1 of this value (*i.e.* very small variation). Thus, the α-pinene SOA H/C ratios reported in this study are consistent with literature values. We did not include the α-pinene SOA H/C and O/C ratios reported in Zhao et al. (2015) in Figure 5 as this would have made the Figure too complex to read (*i.e.* too many data points). We could have estimated the average H/C and O/C ratio from Figure 6A in Zhao et al. (2015) but believed that this did not show the spread in their data and subsequently was not included in Figure 5. We have instead commented on the spread of the data shown in Zhao et al. (2015) in the manuscript (page 15, line 26) which reads, "It is worth noting that a further study, not included in Figure 5 due to the large number of data points, reported H/C ratios ranging between ~ 1.4 to 1.7 for α-pinene SOA which is consistent with the results shown in this work (Zhao et al. (2015))." We have also changed "...very good agreement..." to "...good agreement" throughout the manuscript, *i.e.* "….it can be observed that the H/C and O/C ratios of the SOA samples display good agreement with the literature values….".

**Referee 5**

Pereira et al. presented a chamber operated in continuous mode (CFR) with conceptually the same photochemistry initiating method as the "OFR185" operation mode of oxidation flow reactors (OFRs) (George et al., 2007; Kang et al., 2007; Lambe et al., 2011; Li et al., 2015). They used their CFR to produce very large amounts of SOA for a suite of offline physicochemical analyses, including some requiring high OA amounts. Although the inlet was well controlled and the offline analysis was comprehensive, the reactor and experiment design have a couple of fundamental problems. These are so major that it is unclear to me whether the paper should be published, unless the issues are described thoroughly and the paper used to describe an incremental design step that was not quite successful, which will be built upon to achieve a more atmospherically-relevant reactor in a future iteration.

**Major comments:**

1. Although the volume being larger than OFRs is touted as an advantage of the CFR and part of its novelty, this apparent advantage is negated by the distribution of UV light and species of interest, which appears to be extremely heterogeneous. A pen-ray was used as the light source in the CFR. Although the authors did not specify the dimension of its lighted area in the paper, I searched for this information on the website of its manufacturer (https://www.uvp.com/mercury) and it would appear that it is very small (lighted length as small as < 2 cm). Even if it is slightly larger than that, the pen-ray can be roughly regarded as a point light source given the large volume of the CFR. Then the photon flux scales inversely with the square of the distance to the light source. Let's assume that a spherical UV source with a diameter of 5 cm (much larger surface area than a stick-like lamp with a lighted length of 5 cm). Then there is only 1% of the initial UV intensity (next to the lamp) only ~20 cm from the penray surface under the assumption of no light absorption, and only 0.08% of the initial intensity near the corners of the reactor (assuming the UV light placed right in the middle of the reactor). The UV absorption at 185 nm by $O_2$ exacerbates this problem. $O_2$, with a cross section of ~10-20 cm$^2$ at 185 nm, only needs a ~20 cm optical path to reach an optical depth of 1. This leads to an additional e-fold decay of the intensity every 20 cm, in addition to the intensity decay caused by the geometry. When applying this effect, the light remaining at 20 cm and the reactor corners is 0.6% and 0.0008% of the initial values, respectively. 99.5% (93%) of the reactor volume has light intensities smaller than those near the light by a factor of 10 (100).

Therefore, despite its larger volume than most OFRs, most of the volume in the CFR seems to be photochemically "dark" and its photochemically useful volume is actually smaller than common OFRs, e.g., PAM and CPOT (in almost the whole internal space of PAM (volume: ~13 L) for all commonly used lamp placements, the relative 185 UV intensity to the lamp surface is > 5% (Peng et al., 2018), while for the case discussed above, the volume with a relative 185 nm UV > 5% is only ~3.5 L). Thus, despite its large volume, this is a very small effective reactor.

Even within the photochemically active space, UV intensity still varies substantially, which makes it difficult to relate offline analysis results to a certain reaction condition. Then the results of the offline analysis are less informative.

I suggest that in the future the authors use multiple lights in a better layout to make the UV field as uniform as possible. This would lead to more uniform conditions and the production of more meaningful SOA material.

The authors kindly thank the reviewer for their detailed plots. The longer residence time in the CFR (greater than ~ 25 minutes) will potentially allow more time for the generated SOA to achieve equilibrium with the gas-phase, in comparison to reactors with residence times less than a few hundred seconds (*e.g.* see Anttila et al (2016)). The lighted length of the UV source was 5 cm in experiments 1–32 and 23 cm in experiments 33–38 (the pen-ray was changed to a longer lighted length to increase OH formation in the toluene experiments). We agree that the use of multiple lights would lead to more uniform conditions in the reactor, increasing the atmospheric relevance of the generated SOA. Additional UV lamps could easily be installed in the reactor due to its simple design and will be strongly considered in future design iterations. We must stress however, that the objective of this study was to investigate the effect of chemical composition on the physical state of the generated SOA, furthering our understanding of the physicochemical relationship/s controlling gas-particle partitioning. Recent studies have suggested that particle viscosity is driven by certain chemical components within the aerosol (*e.g.* Huang et al., 2018; Perraud et al., 2012). The comprehensive suite of offline compositional and physical state measurement techniques used in this study (many only accessible because of the large amount of SOA mass which can be generated in the CFR) allows us to further investigate these physiochemical relationship/s. These physiochemical relationship/s are determined by the chemical and physical properties of each SOA sample and are not affected by the atmospheric relevance of generated SOA. We have discussed these comments in the manuscript and have strongly advised in future design iterations that the UV light source is changed.

We have added the following into the manuscript (see page 22, line 11), "The UV lamps used in the CFR had light emissions with wavelengths at 254 nm (primary energy) and 185 nm. The 185 nm wavelength may result in very different organic radical chemistry than observed in the ambient atmosphere (*e.g.* see Peng et al. (2016) for further information), potentially affecting the observed SOA composition. In addition, the light intensity emitted from the UV lamps was not sufficient to provide uniform light distribution within the reactor. It is strongly recommended that the UV light source is modified in future studies, including multiple UV lamps (increasing the light distribution within the reactor) which do not emit a 185 nm wavelength. It must be stressed however, that the objective of this study was to investigate the effect of chemical composition on the physical state of the generated SOA, furthering our understanding of the physicochemical relationship/s controlling gas-particle partitioning. These physiochemical relationship/s are determined by the chemical and physical properties of each SOA sample and are not affected by the atmospheric relevance of generated SOA."

(2) The experiments were conducted using tens of ppm of VOC and several ppm of $NO_x$, corresponding to OH reactivities of thousands of $s^{-1}$ and more (ranging from ~3,000 $s^{-1}$ for Exps. 34 and 35 to ~300,000 $s^{-1}$ for Exps. 26-28). These extraordinarily high reactivities are certain to reduce OH concentration in the CFR by several orders of magnitude (Peng et al., 2015). But UV intensity at 185 and 254 nm is not reduced by the addition of the VOCs, and could play a major role in VOC loss compared to reactions with OH (Peng et al., 2016), especially for toluene, which strongly absorbs at 185 and 254 nm.

Energetic 185 and 254 nm photons may result in a very different organic radical chemistry than in the atmosphere and typical chamber experiments. Although the authors claimed that their objective was not to perfectly mimic atmospheric conditions, clearly a key goal is to produce SOA that is atmospherically-relevant. For example, they repeatedly compared their offline analysis results to ambient measurement and chamber experiment results in the literature as validations of their experiments. If the CFR was only to produce SOA to test several offline analytical instruments with a complex mixture of oxidized species, the current CFR experimental design is purely a laboratory exercise and viable as such. If the authors assume that their CFR-produced SOA may serve as surrogate of ambient and/or typical chamber SOA to any extent (even though the experimental conditions do not replicate ambient conditions), the unrealistic photochemistry initiated by 185 and 254 nm UV should be avoided. In the experiments reported in this paper, OH was always substantially reduced by VOC and $NO_x$. Thus, most of the SOA samples shown in Fig. 5 were only weakly oxidized. Toluene-derived SOA was an exception because strong photolysis at 185 and 254 nm may have produced more organic radicals (followed by $O_2$ addition etc., leading to higher O:C). Those photolysis products were likely to be smaller and more volatile, and have lower SOA yields. The peculiarity of toluene experiments suggests the importance of strong 185 and 254 nm VOC photolysis occurring.

If the authors intend to claim any relevance of CFR-produced SOA to ambient and/or typical chamber SOA, they have to limit the amount of VOC (and $NO_x$) injected to avoid the above problem. OH reactivity of tens of $s^{-1}$ has been recommended for similar reactors using the same OFR185 photochemistry, in order to maintain the chemistry in a tropospherically-relevant regime (Peng et al., 2016; Peng and Jimenez, 2017). This is 100-10000 times lower than the reactivities used in this paper. Assuming a VOC reacting with OH at $10^{-11}$ $cm^3$

molec$^{-1}$ s$^{-1}$, roughly 100 ppb of VOC can be injected into the reactor without entering conditions with significant 185 or 254 nm VOC (and product) photolysis. Then this is roughly the upper limit of OA that can be made in OFRs (including the CFR) using Hg lamps to generate OH. This is about 100 times lower than the concentrations used in this paper.

In this sense, if the authors do not employ multiple lights to largely make use of the volume of the CFR, its SOA production capacity is not superior to other OFRs (e.g., PAM). There have already been a number of papers where other OFRs were used to produce SOA that was deposited on substrates and collected on filters, for SFG and viscosity analysis, respectively (e.g., Shrestha et al., 2015; Song et al., 2016). Besides, PEAR, a large OFR, has been recently presented and appears to have a more appropriate design for the purpose of producing SOA in large quantities (Ihalainen et al., 2018). Thus, I do not think that OFR (CFR) as a tool to produce larger concentrations of SOA (by collecting over a longer period of time) is really a novel concept.

We have not claimed that the SOA generated in the CFR is representative of ambient OA. We have compared the bulk properties (*e.g.* O/C ratio) of our SOA to literature values for information. It is clear that the SOA produced in the CFR has similar O/C ratios as the SOA produced in much low concentrations experiments and therefore is a useful SOA proxy to test analytical tools and the link between composition and physical state (as the reviewer states, a laboratory exercise which is viable as such). The toluene SOA was produced using a longer lighted length UV source (increased to 23 cm), possibly accounting for the higher O:C ratios observed. Nevertheless, we do agree that toluene may have been affected by photolysis from the 185 nm wavelength, as shown in Peng et al. (2016). However, it is important to stress that this does not affect the results of this study (please see our response to major comment 1).

To our knowledge, we use the most comprehensive set of analytical techniques (some which are rarely used) to investigate the chemical composition and physical state of each generated SOA sample. Using this methodology, we are also able to evaluate the accuracy of SOA elemental compositions obtained using ultra-high resolution mass spectrometry (a commonly used technique) and generate non-commercially available standards for SOA quantification. Whilst a number of papers produce SOA and collect onto substrate, there are no papers which use the comprehensive range of analytical techniques employed in this study. For example, Shrestha et al., 2015 did not provide any detailed chemical speciation and Song et al. 2016 provided no compositional data.

The use of "novel" referred to our approach, rather than being solely attributed to the CFR design. This was not clear in our manuscript title and we have therefore changed the title to "A New Aerosol Flow Reactor to Study Secondary Organic Aerosol". Many studies focus on generating SOA using near ambient mixing ratios. This means that many highly accurate analytical techniques cannot be used due to the large amount of SOA mass required for analysis. In this study, we provide a methodology and demonstrate the benefits of using these highly accurate analytical techniques for investigating the complex properties of SOA. The CFR design is however, incredibly versatile and with further modification of the lights as suggested, the CFR could be operated using lower VOC and oxidant mixing ratios, generating more atmospherically relevant SOA. This initial low-cost design can be rapidly modified to suit different research applications. For example, VOC dilution could be achieved with the simple installation of split line. Moreover, a filter could easily be installed on the UV light source to remove 185 nm wavelength within the reactor. In this work, the CFR was used as a simple tool to generate large quantities of SOA mass, although its low cost design (~ £8000) means that the CFR could be used as a cheaper alternative to other reactors (*e.g.* PAM). Furthermore, the high-level of detail provided in the manuscript, allows for easy replication of the CFR design.

It is difficult to directly compare the CFR developed in this study with the PEAR reactor developed by Ihalainen et al. (2018) without further information. Nevertheless, we can make some comparisons. The PEAR reactor has a smaller volume (PEAR 0.14 m$^3$ *vs.* CFR 0.3 m$^3$) and a faster residence time (~ 10 mins *vs.* ~ 25 mins) than the CFR developed in this study and would most likely need to be operated over longer time periods to generate the same amount of SOA mass per experiment. The PEAR reactor is also constructed out of stainless-steel. One of the advantages of using the CFR for this work, is the ability to rapidly replace the reactor sampling bag. Reactors which are constructed out of glass or stainless-steel will require considerable cleaning and are more likely to exhibit "memory effects" from the high mixing ratios (*i.e.* ppmv levels) required for this work. Furthermore, the stainless-steel construction of the PEAR reactor is also likely to cost more to build than the CFR shown in this work.

Whilst we of course endeavour to produce atmospherically relevant SOA, we are also hindered by the large amount of SOA mass required for offline analysis using the highly accurate techniques, such as the

electrodynamic balance, CHNS elemental analyser, nuclear magnetic resonance spectroscopy and semi-preparative liquid chromatography mass spectrometry. To use these analytical techniques, we must use high VOC and oxidant mixing ratios (ppmv levels) to generate a sufficient amount of SOA mass for analysis. Consequently, we are likely to observe higher volatility compounds in the generated SOA than observed under ambient conditions. Only through technological advances (*e.g.* increased instrument sensitivity) or the use of fewer instruments (which was not the purpose of this study) can we overcome this. Please also see our comments to Referee 4, major comment 1 and Referee 3 minor comment 1.

**Specific comments:**

Page 3, Line 30: OFRs (e.g., Aerodyne PAM) also have good flow and precursor injection control.

This sentence has been re-worded and now reads (page 3, line 31), "VOCs and oxidants are continuously introduced into the reactor and sample air extracted, operating under steady-state flow conditions (analogous to oxidative flow reactors), allowing a wide range of chemical scenarios to be investigated through the control of reactant mixing ratios and flow rates (*i.e.* residence time)."

Table 1: there was no really low-$NO_x$ experiments among those with $NO_x$ injected in Table 1. Even with a VOC:$NO_x$ ratio of 13, $NO_x$ was still injected in ppm. Compared to $HO_2$ (not VOC), $NO_x$ should have always dominated $RO_2$ fate in the CFR experiments reported in this paper.

A range of VOC/$NO_x$ ratios were selected and consistently used throughout this study. High mixing ratios were required to generate sufficient quantities of SOA mass. Thus, to achieve the selected VOC/$NO_x$ ratios of 3, 8 and 13, we had to introduce ppmv levels NO to achieve the desired VOC/$NO_x$ ratio. We agree that in the experiments where NO was added, $NO_x$ would have dominated the fate of the $RO_2$ radical. This led to some interesting SOA compositional changes with increasing NO (*i.e.* decreasing VOC/$NO_x$ ratios). We tried to incorporate as many experimental conditions as possible within our time frame, performing 38 experiments in total. However, in future studies, we will allow a sufficient amount of time to incorporate really low-$NO_x$ experiments in our investigations.

Section 2.3: although offline analysis methods are not the main focus of this paper, a brief description of potential artifacts in these offline analyses would still be helpful.

The offline analysis methods are unlikely to introduce artefacts into the samples, providing instrument background or blank runs are regularly performed and the contaminants subtracted from the sample data, as performed in this study. Artefacts are more commonly introduced into the samples through preparation methods (*e.g.* filter extraction processes) for analysis using offline techniques. The use of the electrical low pressure impactor in this work minimised the potential introduction of artefacts into the samples through the exclusion of all extraction processes, *i.e.* samples were either analysed without modification or dissolved into high purity solvent/s. We have included a paragraph on the potential sources of artefacts during sampling, collection and analysis. Please see our response to Referee 3, minor comment 3, which shows our manuscript changes (or page 21, line 27 in the manuscript).

**Technical corrections:**

Figure 1: please change the bag volume from "3 $m^3$" to "0.3 $m^3$".

Changed.

Page 14, Line 5 and Page 19, Line 10: references Cao et al. and Shrivastava et al. are missing in the reference list.

Added.

Page 14, Line 7: please add "of" after "intensity".

Added.

Page 19, Line 33: is "Although a faster evaporation rate…" a part of the preceding sentence?

Yes. This has been changed.

Page 20, Line 10: the word "bin" is missing after "μg m-$^3$".

Added.

Additional references (included in our author response)

Anttila, T., Lehtinen, K. E. J., and Dal Maso, M.: Analytical expression for gas-particle equilibration time scale and its numerical evaluation, Atmospheric Environment, 133, 34-40, 2016.

Reinhardt, A., Emmenegger, C., Gerrits, B., Panse, C., Dommen, J., Baltensperger, U., Zenobi, R., and Kalberer, M.: Ultrahigh Mass Resolution and Accurate Mass Measurements as a Tool To Characterize Oligomers in Secondary Organic Aerosols, Analytical Chemistry, 79, 4074-4082, 2007.

Peng, Z., Day, D. A., Ortega, A. M., Palm, B. B., Hu, W., Stark, H., Li, R., Tsigaridis, K., Brune, W. H., and Jimenez, J. L.: Non-OH chemistry in oxidation flow reactors for the study of atmospheric chemistry systematically examined by modeling, Atmos. Chem. Phys., 16, 4283-4305, 2016.

[revised manuscript text omitted]

---

## Author Response (AR2)

Please see the author response shown in blue below.

Referee 1:
Regarding the UV field, the authors acknowledged that more lamps would greatly help and it would not be difficult to install them. Then the authors should use a design with more lamps as the base CFR design in this study. Or at a minimum, they should clearly acknowledge in the abstract and conclusions that most space in the current CFR design is photochemically dark ("not sufficient to provide uniform light distribution within the reactor" in the revised paper is misleading as a uniform light distribution in reactors is virtually impossible), propose a better lamp layout and discuss its possible improvements compared to the current design.

It is unreasonable to expect the authors to repeat over 2 ½ years of work to increase the number of lights in the reactor, which as previously stated, do not impact the results shown in this work. The referee's comments have evolved from suggesting "in future the authors should use multiple lights in a better layout to make the UV field as uniform as possible" (which we addressed in the manuscript) to repeating the entirety of the study. The referee completely negates the purpose of this study (an approach which we were commended on by Referee 3) and instead focuses on the light source.

We have removed "...not sufficient to provide uniform light distribution within the reactor" and have revised the manuscript text. We have informed readers of the photochemical dark space within the chamber, stressed the importance of UV light modification particularly for applications focused on replicating atmospheric conditions and have suggested a possible lamp layout. We should note, that we have not specified which space within the reactor is photochemically dark (this should be determined from experimental measurements, or modelled using the correct UV lighted lengths). Furthermore, we have stated where we believe multiple UV lamps should be installed to increase the light distribution within the reactor but have kept this discussion brief, as it would require experimental testing. We have not included that the reactor has photochemically dark space in the abstract as it has no impact on the results shown in this work. In addition, "photochemically dark" does not mean "non-reactive". The chamber also contains ozone and potentially $NO_3$ and so further oxidation chemistry may be occurring in these regions. We have however, included this information in the chamber limitations section (where this discussion is most suited) and the conclusions.

Page 22, line 13 reads, "In addition, the light intensity emitted from the UV lamps was not sufficient for the reactor size, resulting in photochemically dark space within the reactor. It is strongly recommended that the UV light source is modified in future studies. This is of the upmost importance for applications focused on replicating atmospheric conditions. Multiple non-emitting 185 nm wavelength UV lamps should be installed into the reactor housing (possibly on all four panels) and the UV emissions experimentally characterised, ensuring the light distribution within the reactor is as uniform as possible, increasing the atmospheric relevance of the generated SOA." Page 23, line 16 reads, "We must stress, that the developed CFR design will require modification to the UV light source for applications focused on replicating atmospheric conditions. The current UV light source is not sufficient for the reactor size, resulting in photochemically dark space within the reactor."

In my comments to the AMTD paper, I already stated that the CFR in the current design is not really an effective setup producing large quantities of SOA. Even under atmospherically irrelevant conditions, PAM and PEAR can still be generally more efficient in producing SOA because their effective photochemically active volumes are larger than that of the CFR. Also, it has been shown that the condensation of gases on OA particles may be kinetically limited in OFRs, but this is not important unless the condensation sink is very low (Palm et al., 2016; Ahlberg et al., 2017; Eluri et al., 2018). A longer residence time of 25 min in the CFR cannot make a major difference in the quantity of SOA produced. Thus, the arguments that the authors used to show that the CFR has advantages in producing large quantities of SOA are not valid.

The novelty of this work does not centre solely on the design of the CFR, but the additional highly accurate compositional and physical state measurements which can be obtained using this methodology – a critical point that the referee continues to ignore. The CFR was not designed to compete with existing highly technical, well established chambers. The CFR was designed as a tool to facilitate the use of advanced analytical instrumentation to gain greater insights into the fundamental physiochemical properties controlling particle dynamics. We do not believe the referee can determine that the longer residence time used in the CFR cannot make a major difference - photo-chemically dark does not mean non-reactive.

The CFR design was successful in generating sufficient quantities of SOA mass, allowing the use of the highly accurate techniques presented. It is therefore irrelevant whether the CFR in its current design, is an effective reactor or not, the main objective of this study was achieved. That said, with further modification of the lights, the

CFR would be a more effective reactor than the PAM and the PEAR at generating SOA. Irrespective, based on its current design, the CFR is a more suitable reactor for this application (discussed in our previous author response).

We have reworded Page 4, line 1 to include the referees comment regarding low condensational sink, removing, "Furthermore, the longer residence times used in the CFR will potentially allow the generated SOA more time to achieve equilibrium with the gas-phase, in comparison to reactors with residences times less than a few hundred seconds (*e.g.* Anttila et al. (2016))". Replacing with, "Furthermore, studies have shown that the oxidation of some precursors with low condensational sinks may not achieve gas-particle equilibrium in reactors with residence times less than a few hundred seconds (Ahlberg et al., 2017; Anttila et al., 2016; Eluri et al., 2018; Palm et al., 2016). The longer residence times used in the CFR could potentially be used to overcome this, allowing the generated SOA more time to achieve equilibrium with the gas-phase".

The authors acknowledged that they did not aim to use the CFR to produce atmospherically relevant SOA. Then the scientific significance of this paper is seriously limited. While I agree that the relationship between the chemical composition and physicochemical properties of SOA is not affected by its atmospheric relevance, a reactor that can only explore this relationship in a part of the composition/property space that is far from atmospherically relevance has very limited usefulness. As I said in my review of the AMTD paper, the conditions under which the CFR were run were remarkably different (orders of magnitude away) from the atmospherically relevant conditions. To my knowledge, there is not any easy solution that can efficiently produce mg of SOA that can serve as ambient SOA surrogate. If the authors can use the CFR to achieve this goal, that will be a real novelty and should be detailed in this paper. Otherwise, the authors should clearly acknowledge that the CFR is unable to produce ambient surrogate SOA when producing SOA in large quantities.

We did not state that we did not aim to produce atmospherically relevant SOA. We stated that "our objective was not to mimic atmospheric conditions", simply because it was not possible. We later stated in our author response, "whilst we of course endeavour to produce atmospherically relevant SOA, we are hindered by the large amount of SOA mass required for offline analysis using the highly accurate techniques". The use of these highly accurate techniques is the fundamental concept of this study. We must use high VOC and oxidant mixing ratios to generate sufficient SOA mass to use these techniques. Techniques which are usually inaccessible (due to large amount of SOA mass required), allowing us to further probe the complex physiochemical properties controlling SOA dynamics.

The CFR is able to produce surrogate SOA in large quantities. We have shown that the bulk elemental composition of the SOA is in good agreement with literature values and is consistent with the characteristic Van Krevelen diagram trajectory (as shown in Figure 5). In addition, we have identified numerous well known α-pinene oxidation products in the generated SOA (Figure 7 and SI Table S2). Surrogate, by definition, means a substitute. Our results show that the bulk chemical composition of the generated SOA is largely unaffected by the use of high mixing ratios and conditions (as mentioned above) and thus, we argue the generated SOA is a good surrogate. Indeed the two previous reactor papers highlighted above (Ahlberg et al., Lambe et al.) both present their compositional data as O:C ratios from AMS and do not actually investigate the detailed composition. There are numerous high profile papers that have used surrogate SOA material which is far less representative than the generated SOA reported in this study. For example Pfrang et al., 2017 (Nature Communications) study 3-dimensional self-assembly in aerosol proxies using sodium oleate/oleic acid solutions and simple fatty acid/sugar/hydrocarbon mixtures. These simple mixtures cannot mimic atmospheric aerosols where many organic species are present at low concentrations, and our SOA proxy material could be used as an intermediate step, with better characterised composition, to extend the results to more atmospherically relevant mixtures. Therefore we strongly refute the claim that our study is scientifically limited.

Other comments:
Response 1 to Referee 4: if experiments are run with ppmv of precursors, they are very likely atmospherically irrelevant. Then the only goal of these experiments is to make as much SOA as possible. It is unclear to me why "memory effects" should be considered as a problem. In other words, I am wondering why "memory effects" should be taken more seriously than atmospheric irrelevance, as to me both are types of inabilities to explore certain parts of SOA chemical composition space.

There are several points here. Firstly, the generated SOA is not atmospherically irrelevant. Our results show that the bulk chemical composition is largely unaffected by the use of high VOC and oxidant mixing ratios, displaying good agreement with literature values and are consistent with the characteristic Van Krevelen diagram trajectory (shown in Figure 5). We have also identified numerous well known α-pinene oxidation products in the generated SOA samples (Figure 7 and SI Table S2) and thus, our samples are not atmospherically irrelevant.

Second, the above comments read (in our opinion), if you cannot achieve your ultimate goal (*i.e.* complete atmospheric relevance) why perform other parts of the study correctly? Memory effects or more specifically, reactor contamination from previous experiments, should absolutely be considered in the reactor design. We must use high VOC and oxidant mixing ratios to generate sufficient SOA mass to allow the use of the highly accurate state-of-the-art instrumentation presented in this study, techniques which are rarely used. We cannot overcome the use of high VOC and oxidant mixing ratios unless the instrumentation advances (*i.e.* increased sensitivity). The goal is not to generate as much atmospherically irrelevant material as possible, but to generate sufficient quantities of SOA mass which is as atmospherically relevant as possible.

Besides, I do not think that a custom-built PAM itself is expensive.

We did not state that the PAM reactor was expensive. We stated that the CFR is a cheaper alternative, which it is.

Response to my "Table 1" comment: I do not understand this response. If the authors wished to conduct low-NOx experiments, simply injecting no NO would work, since NO is not necessary for photochemical production of OH in the CFR.

We do not understand this comment. We did perform experiments without NO as clearly stated in our experimental section.

[revised manuscript text omitted]